# Structural basis of thiamine transport and drug recognition by SLC19A3

Florian Gabriel [1,2], Lea Spriestersbach [1,2], Antonia Fuhrmann [1,2], Katharina E. J. Jungnickel [1,2], Siavash Mostafavi [1,2], Els Pardon [3,4], Jan Steyaert [3,4] & Christian Löw [1,2,5] ✉

Thiamine (vitamin $B_1$) functions as an essential coenzyme in cells. Humans and other mammals cannot synthesise this vitamin de novo and thus have to take it up from their diet. Eventually, every cell needs to import thiamine across its plasma membrane, which is mainly mediated by the two specific thiamine transporters SLC19A2 and SLC19A3. Loss of function mutations in either of these transporters lead to detrimental, life-threatening metabolic disorders. SLC19A3 is furthermore a major site of drug interactions. Many medications, including antidepressants, antibiotics and chemotherapeutics are known to inhibit this transporter, with potentially fatal consequences for patients. Despite a thorough functional characterisation over the past two decades, the structural basis of its transport mechanism and drug interactions has remained elusive. Here, we report seven cryo-electron microscopy (cryo-EM) structures of the human thiamine transporter SLC19A3 in complex with various ligands. Conformation-specific nanobodies enable us to capture different states of SLC19A3's transport cycle, revealing the molecular details of thiamine recognition and transport. We identify seven previously unknown drug interactions of SLC19A3 and present structures of the transporter in complex with the inhibitors fedratinib, amprolium and hydroxychloroquine. These data allow us to develop an understanding of the transport mechanism and ligand recognition of SLC19A3.

Thiamine, commonly known as vitamin $B_1$, is crucial for the survival of cells and an essential micronutrient for humans and other mammals[1]. Inside the cell, thiamine is converted to thiamine pyrophosphate (thiamine-pp) by the enzyme thiamine pyrophosphokinase 1 (TPK1)[2,3]. Thiamine-pp then acts as a coenzyme in central metabolic pathways, such as the citric acid cycle and the pentose phosphate pathway[4]. Mammalian cells lack the ability to synthesise thiamine de novo. Instead, they import the vitamin over their plasma membrane through the transport activity of several integral membrane proteins of the solute carrier family (SLC, Fig. 1a)[5]. SLC19A2 and SLC19A3 are the main transporters for thiamine in humans and mediate the uptake of the vitamin with high affinity and specificity ($K_m = 2-7\,\mu M$)[6–8]. They are closely related to the folate transporter SLC19A1 (35% sequence identity, 48% sequence similarity), with which they form the SLC19A family of vitamin transporters[5]. In addition, the organic cation transporter 1 (OCT1, SLC22A1), which is known to accept a wide variety of substrates, can also mediate thiamine uptake, though with lower affinity ($K_m$-780 $\mu M$)[9]. Among the known thiamine transporters, SLC19A3 stands out in terms of its physiological and pharmacological importance, as it is crucial for the transport of thiamine across the intestinal wall and the blood-brain barrier[10,11]. Deletion of SLC19A3 in mice leads to strongly decreased thiamine levels in the blood serum and brain tissue

[1]Centre for Structural Systems Biology (CSSB), Notkestraße 85, 22607 Hamburg, Germany. [2]European Molecular Biology Laboratory (EMBL) Hamburg, Notkestraße 85, 22607 Hamburg, Germany. [3]Structural Biology Brussels, Vrije Universiteit Brussel (VUB), 1050 Brussels, Belgium. [4]VIB-VUB Center for Structural Biology, VIB, 1050 Brussels, Belgium. [5]Bernhard Nocht Institute for Tropical Medicine, 20359 Hamburg, Germany. ✉e-mail: christian.loew@embl-hamburg.de

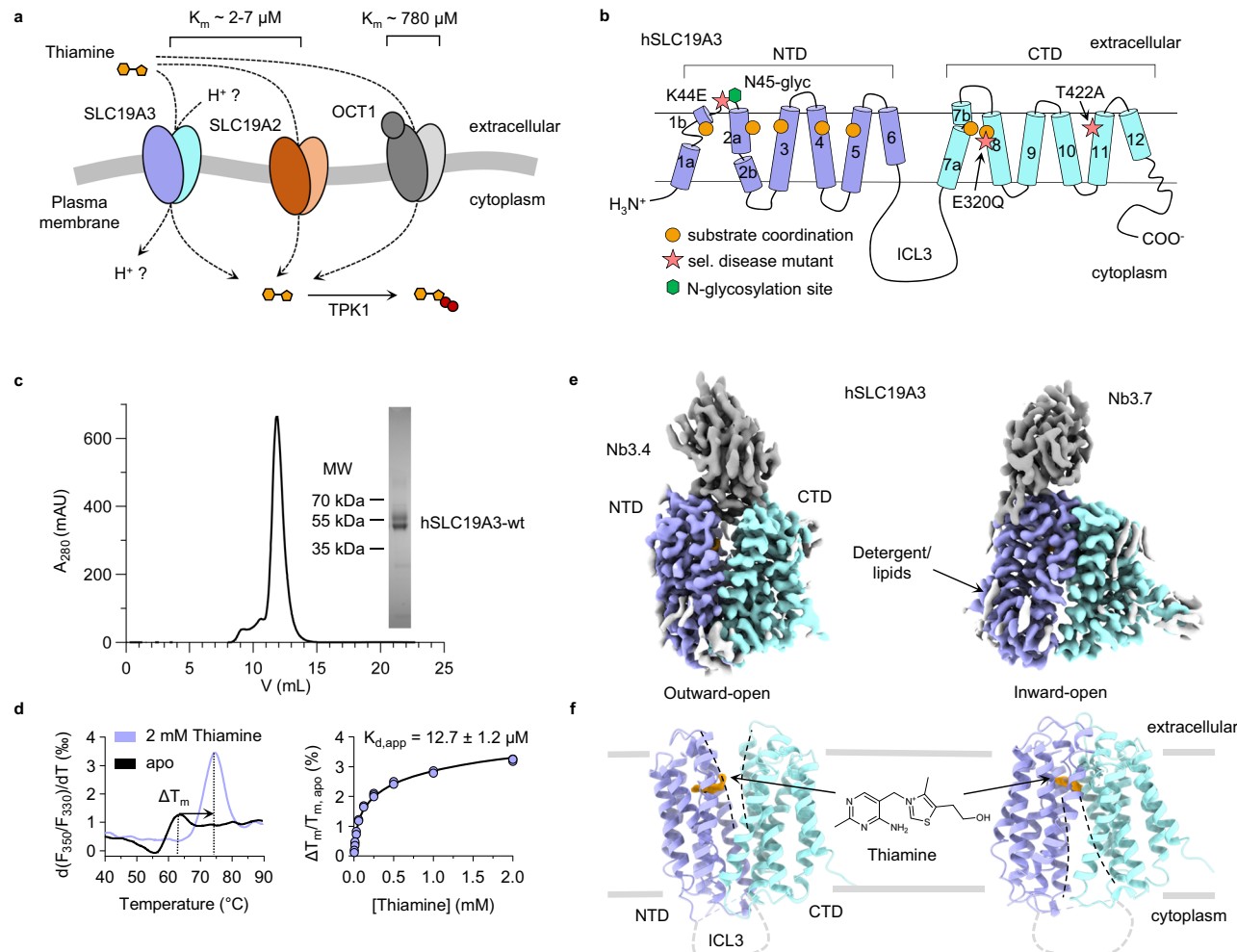

**Fig. 1 | Structure and function of hSLC19A3. a** Humans and other mammals have at least three distinct membrane transporters that can mediate thiamine uptake: SLC19A3, SLC19A2 and OCT1 (SLC22A1). SLC19A3 is essential for the uptake of thiamine across the intestinal and blood-brain barrier under physiological conditions. In the cytoplasm, thiamine is phosphorylated by the enzyme TPK1 to form the biologically active coenzyme thiamine pyrophosphate. **b** Cartoon representation of hSLC19A3, based on the determined cryo-EM structures. The transporter follows the canonical MFS fold, with twelve transmembrane helices (TMs) folding in two symmetrically related six-helix bundle domains (NTD and CTD). Substrate coordination is mediated mostly by residues of TM1-5 of the NTD and by TM7 and TM8 of the CTD (orange circles). The positions of three disease mutants which were studied in this work are indicated by pink stars. **c** Representative size-exclusion chromatography (SEC) trace of purified hSLC19A3-wt. The protein elutes as a monomer of about 50 kDa. A fraction of the protein is glycosylated and appears as an extra band at ~60 kDa in SDS-PAGE (inset). The purification of hSLC19A3-wt has

been performed >10 times with close to identical outcomes. **d** Thermal shift assay of hSLC19A3 using nanoDSF. The left panel shows the first derivative of the melting curve, measured as ratio of the fluorescence recorded at 350 nm (F350) and at 330 nm (F330), in the absence and presence of thiamine. Thiamine induces a strong stabilisation of the transporter ($\Delta T_m = 10.9 \pm 0.3$ °C by 2 mM thiamine). The right panel illustrates the concentration dependent thermal shifts ($n = 3$) with the resulting apparent dissociation constant $K_{d,app}$ for thiamine (mean ± s.d.). **e** Cryo-EM maps of thiamine-bound hSLC19A3 in its outward-open (left) and inward-open (right) conformation. The NTD (purple) and CTD (cyan) of the transporter, as well as the fiducial nanobodies (grey), are resolved. ChimeraX contour levels: outward-open: 0.876; inward-open: 0.148. **f** Structure models of the respective conformational states. The density of thiamine is shown in orange. ChimeraX contour levels: outward-open: 0.547; inward-open: 0.148. The N-terminus, C-terminus and ICL3 are expected to be structurally disordered and could consequently not be resolved.

under standard diet conditions[11–13]. Mutations of SLC19A3 in humans cause severe neurometabolic disorders in the form of Wernicke's-like encephalopathy (WLE)[14] and biotin- and thiamine-responsive basal ganglia disease (BTBGD)[15–17]. In most cases, these diseases have an onset in infancy and can lead to life-long disabilities and early death[15]. Thiamine uptake through SLC19A3 is further known to be inhibited by a broad spectrum of commonly prescribed drugs, including antidepressants, antibiotics and antineoplastic medications[7,10]. Being under treatment with these thiamine uptake inhibitors (TUIs) can lead to drug-induced thiamine deficiencies on an organism-wide or tissue-specific level, with potentially fatal consequences[11]. This is exemplified by the case of the Janus-kinase (JAK) inhibitor fedratinib. Treatment with this compound induced the development of Wernicke's encephalopathy, a severe neurodegenerative disorder characteristic for thiamine

deficiency, in several patients[18] (clinical trial ID: NCT01437787). This adverse event could eventually be linked to high-affinity inhibition of SLC19A3 by fedratinib[19].

The human SLC19A3 (hSLC19A3) transporter is a twelve-pass transmembrane protein of the Major Facilitator Superfamily (MFS)[20–22]. It is predicted to follow the canonical MFS fold, which consists of two α-helical and symmetrically related domains, termed the N-terminal (NTD) and the C-terminal domain (CTD), as shown in Fig. 1b. In recent years, the transport activity and drug-induced inhibition of SLC19A3 has been thoroughly studied on a cellular level[7,8,10,23–27]. The molecular basis of ligand recognition and the transport mechanism of SLC19A3 were, however, for a long time not well understood.

Here, we report seven cryo-EM structures of hSLC19A3. Structure determination is enabled by the generation of three conformation-

specific nanobodies against hSLC19A3. With these, we capture the solute carrier in its outward- and inward-open states, which provides detailed insights in its substrate recognition and transport cycle. Furthermore, we use thermal shift assays in conjunction with cellular thiamine-uptake assays to screen for compounds that inhibit thiamine transport through hSLC19A3. This brings about the discovery of seven previously unknown drug interactions of the transporter. We further present cryo-EM structures of the transporter in complex with the known high-affinity inhibitors fedratinib, amprolium and hydroxychloroquine, which elucidates the structural basis of drug interactions of hSLC19A3.

## Results

### Cryo-EM structures of hSLC19A3 in different conformations

To study hSLC19A3 on a structural and biophysical level we made use of an expression system that has already been widely used for the functional characterisation of the transporter in cell-based assays[7,8,10,24]. Full-length hSLC19A3 was expressed in HEK293-derived Expi293F™ cells and purified in a detergent solution using a mixture of lauryl maltose neopentyl glycol (LMNG) and cholesterol hemisuccinate (CHS) (Fig. 1c). The recombinant transporter is strongly stabilised against heat denaturation in the presence of its known substrate thiamine ($\Delta T_m = 10.9 \pm 0.3\,°C$, Fig. 1d). Concentration-dependent thermal shift assays revealed high-affinity binding of thiamine with an apparent dissociation constant ($K_{d,app}$) of $12.7 \pm 1.2\,\mu M$ (Fig. 1), which is in good agreement with previously reported transport affinities in the low micromolar range[8,9]. In contrast, phosphorylation of thiamine strongly reduced the interaction between the vitamin and hSLC19A3 (Supplementary Fig. 1a). Since hSLC19A3 on its own proved to be too small for structure determination by cryo-EM (~55 kDa), we generated hSLC19A3-specific nanobodies in a llama (Supplementary Fig. 2). For immunisation, a glycosylation-free mutant of hSLC19A3 (hSLC19A3-gf) was used in order to provide an increased accessible surface area for antibody binding (Supplementary Fig. 3). To generate hSLC19A3-gf, the predicted N-glycosylation sites Asn45 and Asn166 were mutated to glutamine residues. Thermal stability experiments did not show any interference of these mutations with thiamine binding (Supplementary Fig. 4). The immunisation and nanobody discovery were performed as described earlier[28]. In total we identified three hSLC19A3-specific binders: Nb3.3, Nb3.4 and Nb3.7 with binding affinities in the range of 100-300 nM (Supplementary Fig. 2). While Nb3.3 and Nb3.4 bind stably to wildtype hSLC19A3, Nb3.7 can sterically only interact with the glycosylation-free transporter (Supplementary Fig. 3b). All three nanobodies were used for structure determination of hSLC19A3 by single particle cryo-EM with map resolutions reaching between 2.9-3.8 Å on a global level and 2.4-3.2 Å in the substrate binding site (for an overview, see Supplementary table 1). Nb3.4 stabilises the outward-open state of the transporter, while Nb3.3 and Nb3.7 both bind to its inward-open state (Fig. 1e,f, Supplementary Fig. 6-10). Using this molecular toolset, we determined the structures of hSLC19A3 in its outward-open and inward-open conformations, in both apo and thiamine-bound states.

The experimental structures of hSLC19A3 confirm the MFS fold of the transporter, with twelve transmembrane helices (TM) folding into two compact six-helix bundle domains (Fig. 1f, Supplementary Fig. 11). The domains are connected via a 76-residues long (Lys195-Glu271), poorly conserved intracellular loop 3 (ICL3) (Supplementary Fig. 12). No density could be resolved for this loop, indicating that this part of the transporter is likely disordered and flexible. Similarly, parts of the N-terminus (Met1-Ser10) and the C-terminus (Tyr459-Leu496) could not be resolved in any of the cryo-EM structures (Fig. 1e, f). Within the well resolved helical domains, there are three discontinuous transmembrane helices in both the outward- and inward-open state: TM1, TM2, and TM7. In TM1, Ile13-Met27 form a stable α-helix (TM1a). The helix then partially unwinds and is followed by a short helical segment

(TM1b), which is flanked by two proline residues in position 33 and 42 (Supplementary Fig. 11a). TM2 is interrupted close to its cytoplasmic end, between Val65 and Leu68. This discontinuity allows TM2 to bend around TM4 (Supplementary Fig. 11a). The helical structure of TM7 is also broken, in this case close to the extracellular space, between Asn297 and Gln300 (Supplementary Fig. 11b). Our structural data further confirm that recombinant hSLC19A3 is N-glycosylated on Asn45, whereas the second predicted glycosylation side, Asn166, appears to be glycosylation-free in the wildtype transporter (Supplementary Fig. 3). Additional non-protein density observed in the EM-reconstructions likely originates from bound lipids or detergent molecules (Fig. 1e).

### Thiamine binding site

Comparison of the final cryo-EM reconstructions from data sets recorded in the absence and presence of thiamine revealed the binding mode of the vitamin to hSLC19A3. In the thiamine-free samples, the central vestibule of the transporter retains some weak densities (Supplementary Fig. 13). These originate most likely from small molecules of cellular origin, buffer components or coordinated water molecules. This observation is not uncommon in cryo-EM maps[29]. In the thiamine-supplied samples, in contrast, extra densities appeared in the vestibule of hSLC19A3, which correspond well to the molecular shape of thiamine. The fitted ligand also matches with the transporter structures in terms of coordinating interactions (Fig. 2, Supplementary Fig. 13). These structures highlight that the substrate binding site is positioned close to the extracellular side of hSLC19A3 (Fig. 2a, b). Thus, the thiamine binding site in hSLC19A3 is similar to the reported substrate binding pocket of the closely related folate transporter SLC19A1[30–32] (Supplementary Fig. 14b, d, f; the corresponding binding site residues are also conserved in hSLC19A2, Supplementary Fig. 15). This is, however, a rather unusual feature for MFS transporters, which typically bind their substrates more centrally within the membrane plane[33] (Supplementary Fig. 16). Since the discovered nanobodies allowed the determination of different conformational states of hSLC19A3, a direct comparison of thiamine binding in the outward-open and the inward-open state was possible.

In the outward-open state, thiamine is exclusively coordinated by the N-terminal domain (NTD) of the transporter (Fig. 2a, c). Thiamine adapts a kinked conformation with regard to its two aromatic rings. Its aminopyrimidine moiety inserts into a complementary pocket of the NTD, parallel to the membrane plane (Fig. 2a, c). Here, the aromatic ring interacts via π-π stacking with Tyr113 and to a lesser extend with Trp59 (Fig. 2c, e). The methyl group of the aminopyrimidine ring extends into a hydrophobic pocket formed by the side chains of Thr93, Trp94 and Leu97 on TM3, and Met106 and Val109 on TM4 (Fig. 2e). Crucial polar contacts are formed with Glu110, which is within hydrogen bonding distance ($\leq 3.0\,\text{Å}$)[34] to both the ring-nitrogen (2.8 Å), as well as the primary amine of thiamine (2.8 Å) (Fig. 2e). The latter ones constitute a hydrogen bond acceptor and donor, respectively. We hypothesise that Glu110 could be protonated in the thiamine-bound state to provide a matching hydrogen bond donor-acceptor pair for the aminopyrimidine ring (Fig. 2e, Supplementary Fig. 1c, d). Structure-based $pK_a$ estimations using PROPKA[35] rank the $pK_a$ of Glu110 in the neutral range ($pK_a = 6.4$–$7.5$), which would support the idea of at least partial protonation of this residue under physiological conditions. The specific hydrogen bonding between the aminopyrimidine ring and Glu110 appears to be critical for the high-affinity binding of thiamine to hSLC19A3. In oxythiamine, a close derivative of thiamine, the primary amine of the aminopyrimidine ring is substituted by a carbonyl oxygen. This subtle change already leads to a strongly decreased affinity and a lower thermostabilisation of the transporter (Supplementary Fig. 1a, d). A similar effect is observed, when Glu110 is mutated to glutamine, resulting eventually in a drop of binding affinity from an apparent $K_d$ of $12.7 \pm 1.2\,\mu M$ to $85 \pm 13\,\mu M$.

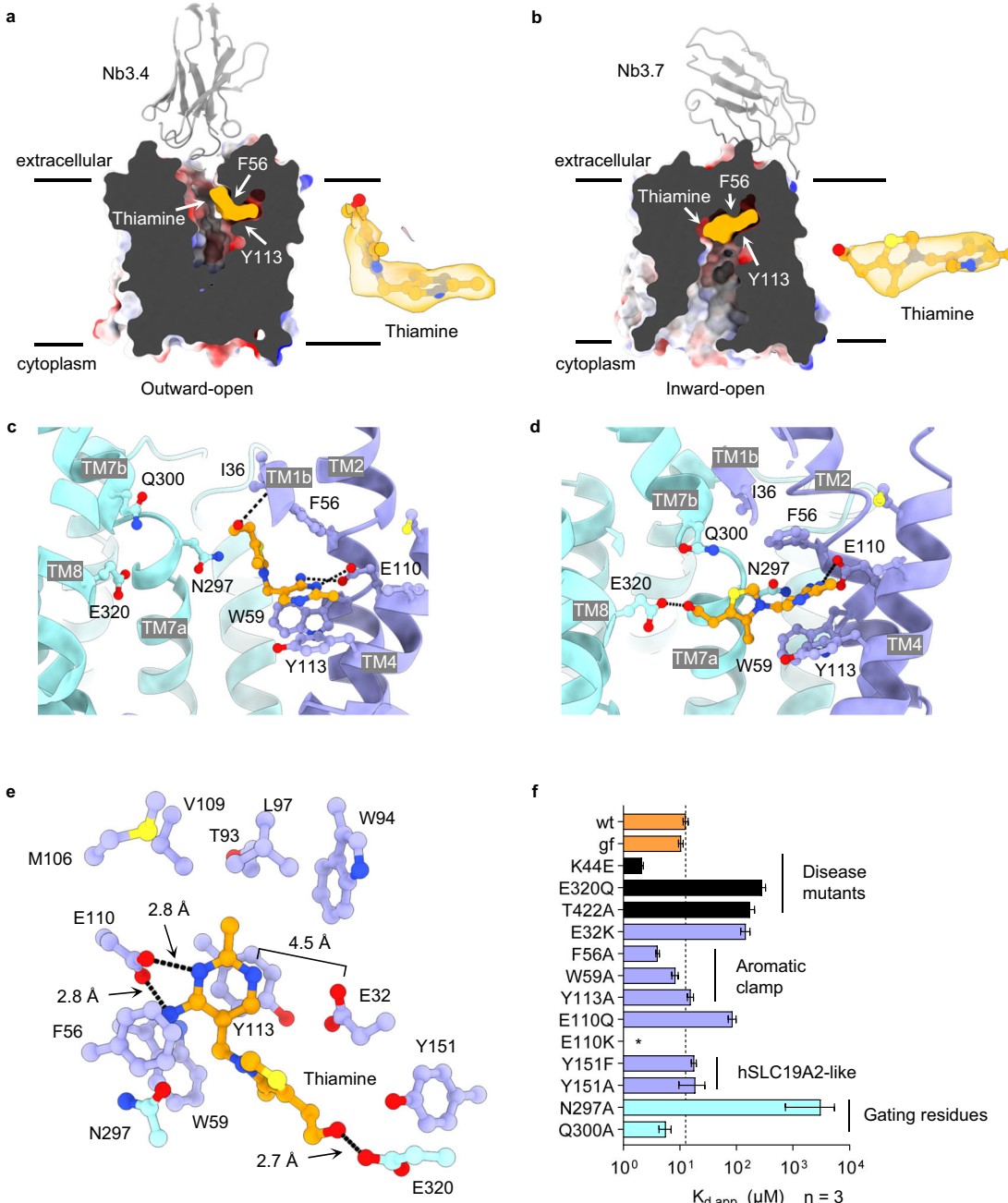

**Fig. 2 | Thiamine binding site of hSLC19A3. a** Localisation of the thiamine binding site of hSLC19A3 in the outward- and (**b**), inward-open conformation of the transporter. The transporter is shown in electrostatic surface representation. Density for thiamine (orange) is shown separately (ChimeraX contour levels: 0.547 and 0.148, Q-scores: 0.45 and 0.57). **c** Residues involved in thiamine coordination in the outward- and (**d**), inward-open conformation of hLSC19A3, shown in the membrane plane. **e** Extracellular view of hSLC19A3 highlighting the thiamine-coordinating residues in the substrate binding site of the transporter. Residues of the NTD are coloured in purple, residues of the CTD in cyan, and thiamine in orange. Black dashes indicate hydrogen bonds (cut-off at 3.0 Å)[33] **f** Thiamine binding affinities (apparent dissociation constants $K_{d,app}$) of single point mutants of hSLC19A3. The corresponding titration curves were measured in three independent experiments ($n = 3$, see Supplementary Fig. 4). The bar plot shows the mean $K_{d,app}$ and its standard deviation derived through a global fit of the measured data to Eq. (1), as described in the methods. For the purification of the respective mutants, see Supplementary Fig. 20.

The replacement of Glu110 by a positively charged lysine residue renders hSLC19A3 completely unresponsive to thiamine (Supplementary Fig. 5). Opposite of Glu110, Glu32 provides another polar contact, which is coordinating the aminopyrimidine ring of thiamine and has been shown to be important for the transport activity of hSLC19A3[36,37] (Fig. 2e). In the outward-open state of hSLC19A3, the thiazolium ring of thiamine is oriented approximately 90° relative to the aminopyrimidine ring and is primarily coordinated through π-π stacking with Phe56. The hydroxyethyl group, the tail of thiamine,

reaches even further up in the binding site, forming polar contacts with the backbone of Ile36 (Fig. 2c).

When hSLC19A3 transitions from the outward- to the inward-open state, thiamine undergoes a major rearrangement. It relaxes via its rotatable *C-C* and *C-N* bonds connecting the aminopyrimidine and thiazolium ring into an elongated conformation (Fig. 2d, e). The aminopyrimidine ring stays coordinated in the same position in the NTD, with Phe56 closing on top of it. Together with Trp59 and Tyr113, Phe56 forms now a hydrophobic pocket that we termed 'aromatic clamp'

(Fig. 2e). Meanwhile, the positively charged thiazolium ring reorients and is accommodated between the electronegative backbone and side chain carbonyl oxygen atoms of Leu296 and Asn297, respectively. Through the transition of hSLC19A3 to the inward-open conformation, the hydroxyethyl tail of thiamine detaches from the backbone of Ile36 and reaches across to Glu320 on the CTD to form a hydrogen bond (2.7 Å). With this, thiamine tightly connects the two helical bundles of the transporter by non-covalent interactions (Fig. 2d, e). Glu320 is further predicted to undergo strong $pK_a$ changes over the course of the transport cycle, which could potentially link thiamine transport to a proton gradient over the membrane (see discussion).

In order to obtain experimental insights into the interaction of individual residues with thiamine, we generated and analysed several single point mutants of hSLC19A3 (Fig. 2f, Supplementary Fig. 4, 20). The mutation of residues of the aromatic clamp to alanine did not impair thiamine binding (Fig. 2f). We speculate that two out of the three constituents of the aromatic clamp (Phe56, Trp59 and Tyr113) are sufficient to mediate stable π-π stacking of the aminopyrimidine ring of thiamine. In contrast, mutating Asn297 to alanine, and thereby removing a negative partial charge from the substrate binding site, led to a -250-fold loss in binding affinity (Fig. 2f). In addition, we probed the role of Glu32 in thiamine binding. The mutation of the equivalent residue Glu45 to lysine in the closely related folate transporter SLC19A1 leads to a loss of function and conveys methotrexate (MTX) resistance[38,39] (Supplementary Fig., 14). The analogous mutation (hSLC19A3-Glu32Lys) causes a 10-fold drop in the affinity for thiamine in hSLC19A3, from $12.7 \pm 1.2\,\mu M$ to $145 \pm 27\,\mu M$. It is likely that the placement of a positive charge in this position leads to an electrostatic or partially steric repulsion of the aminopyrimidine ring (Fig. 2e, f). The interaction between Glu320 and the hydroxyethyl tail of thiamine is affected in a rare form of heritable Wernicke's-like encephalopathy[14]. Our biophysical data illustrate that the disease-associated mutation Glu320Gln leads to a 20-fold decrease in affinity for thiamine, from $12.7 \pm 1.2\,\mu M$ to $284 \pm 45\,\mu M$ (Fig. 2f), in agreement with recently published transport data[36].

### Substrate binding sites in other human thiamine transporters

The transport of thiamine can be mediated by different transporters, dependent on the tissue, cell type and subcellular compartment[23,25,40,41] (Fig. 1a). High-affinity transport of thiamine across the plasma membrane is shared between SLC19A3 and the closely related SLC19A2[6,26]. In order to understand the structural relationship between these two, we compared the experimentally determined structure of the inward-open hSLC19A3 with the AlphaFold2 (AF2) predicted structure of hSLC19A2[42] (Supplementary Figs. 14, 15).

The global fold of the two proteins is almost identical, with the only marked differences in the poorly conserved and disordered N- and C-terminal regions, as well as ICL3 (Supplementary Fig. 14b). The substrate binding site appears to be structurally highly conserved. Only subtle differences in the form of two substitutions, (i) Phe56[hSLC19A3] to Tyr74[hSLC19A2] and (ii) Tyr151[hSLC19A3] to Phe169[hSLC19A2] can be observed (Supplementary Fig. 14e). The first substitution introduces a minor change in the aromatic clamp. As shown in Fig. 2f, replacing Phe56[hSLC19A3] with alanine in hSLC19A3 did not impact the binding affinity for thiamine. Therefore, we assume that the even more similar substitution of Phe56 to tyrosine will not impact substrate binding. The effect of the second substitution (ii) on thiamine binding was tested experimentally. When Tyr151[hSLC19A3] was replaced with phenylalanine, no strong effect on the affinity for thiamine was observed (Fig. 2f). This is in line with previous observations, which reported very similar transport affinities for the two thiamine transporters[7,8]. The situation is different, when comparing the hSLC19A3 structures with the recently determined cryo-EM structure of OCT1 (SLC22A1)[29] in complex with thiamine. The reported transport affinity of OCT1 for thiamine is much lower than for the SLC19A transporters ($K_m$ ~780 μM compared to 2–7 μM)[7–9]. This

difference could be explained by a looser coordination of thiamine in OCT1, compared to the tight binding observed in hSLC19A3[29] (Supplementary Fig. 16).

### Transport cycle

To obtain a better understanding of the conformational changes during the transport cycle we compared the cryo-EM structures of hSLC19A3 stabilised in the outward-open state by Nb3.4 with the inward-open state, bound by Nb3.3 or Nb3.7. An overlay of the individual N- and C-terminal domains of the different conformational states revealed no major intradomain rearrangements ($C_\alpha$ RMSD of 0.52–1.94 Å). Only extracellular loop 4 (ECL4) and intracellular loop 5 (ICL5) reorient slightly (Supplementary Fig. 11). Based on the available structures, we conclude that the conformational changes observed for hSLC19A3 follow a rocker-switch mechanism, in which the NTD and CTD move as rigid bodies to create a moving barrier around the substrate binding site (Supplementary Movies 1–3)[33]. This barrier consists of a cytoplasmic gate and an extracellular gate that open and close in an alternating fashion. This provides alternating solvent accessibility to the substrate binding site from the extracellular and cytoplasmic space (Supplementary Fig. 17). The cytoplasmic gate of hSLC19A3 is formed by hydrophobic contacts of Tyr128 (TM4) and Ala395 (TM10) with Tyr403 (TM11). The gate is further stabilised by polar interactions of the side chains of Asp75 (TM2) and Gln137 (TM5) with the backbone of Ala404 and Leu405 (TM11) and Lys338 (ICL5), respectively (Supplementary Fig. 17a–c). These contacts are broken when the transporter transitions to the inward-open state (Supplementary Fig. 17d–f). The long ICL3 of hSLC19A3 allows for a dilation of the cytoplasmic gate. Simultaneously, a barrier to the extracellular space is established by the closure of the extracellular gate. The core of this gate is built by polar contacts of the side chains of Asn297 and Gln300 (TM7). They are within hydrogen bonding distance with the backbone of Phe56 (TM2) and Pro33 (TM1a), respectively. These polar contacts are supported by hydrophobic interactions between Ile36 (TM1b) and Ile301 (TM7). With these in place, TM1b is coordinated as a lid, sealing the extracellular gate (Supplementary Fig. 17d–f).

### Mutation of hSLC19A3 in biotin- and thiamine responsive basal ganglia disease (BTBGD)

Loss of function mutations of hSLC19A3 cause BTBGD in humans[15,17,43–47] (Supplementary Fig. 18, 19). This disease is marked by severe neurological symptoms and is inherited genetically in an autosomal recessive manner[15]. The median age of onset of symptoms varies between three months and four years after birth, dependent on the underlying mutations in the *SLC19A3* gene. Under the currently advised treatment - a combination therapy of high-dose thiamine and biotin - more than 30% of the patients still suffer from moderate to severe effects of the disease, while about 5% of those affected die despite treatment[15]. To gain a better understanding of the molecular cause of this disorder, we investigated the effects of one of the most common BTBGD-causing point mutation SLC19A3-Thr422Ala. Most cases are reported in Saudi Arabia, where the heterozygous carrier frequency is about 1 in 500 among the Saudi population[17]. The patients usually present with symptoms including subacute encephalopathy with confusion, convulsions, dysarthria and dystonia[48]. On a cellular level, the SLC19A3-Thr422Ala mutant localises normally to the (apical) plasma membrane in Caco-2 and MDCK cells[25] and can also be expressed and purified in vitro (Supplementary Fig. 20a). However, its thiamine uptake activity is significantly impaired[25]. This agrees with our data, highlighting that the Thr422Ala mutation leads to a more than 10-fold decrease of binding affinity for thiamine, from $12.7 \pm 1.2\,\mu M$ to $172 \pm 35\,\mu M$ (Fig. 2f, Supplementary Fig. 19d). Thr422 is located on TM11 and thus spatially in the outer region of the transporter. It is, however, physically connected to the core of hSLC19A3 by hydrogen bonding with the side chain of Gln294 on TM7

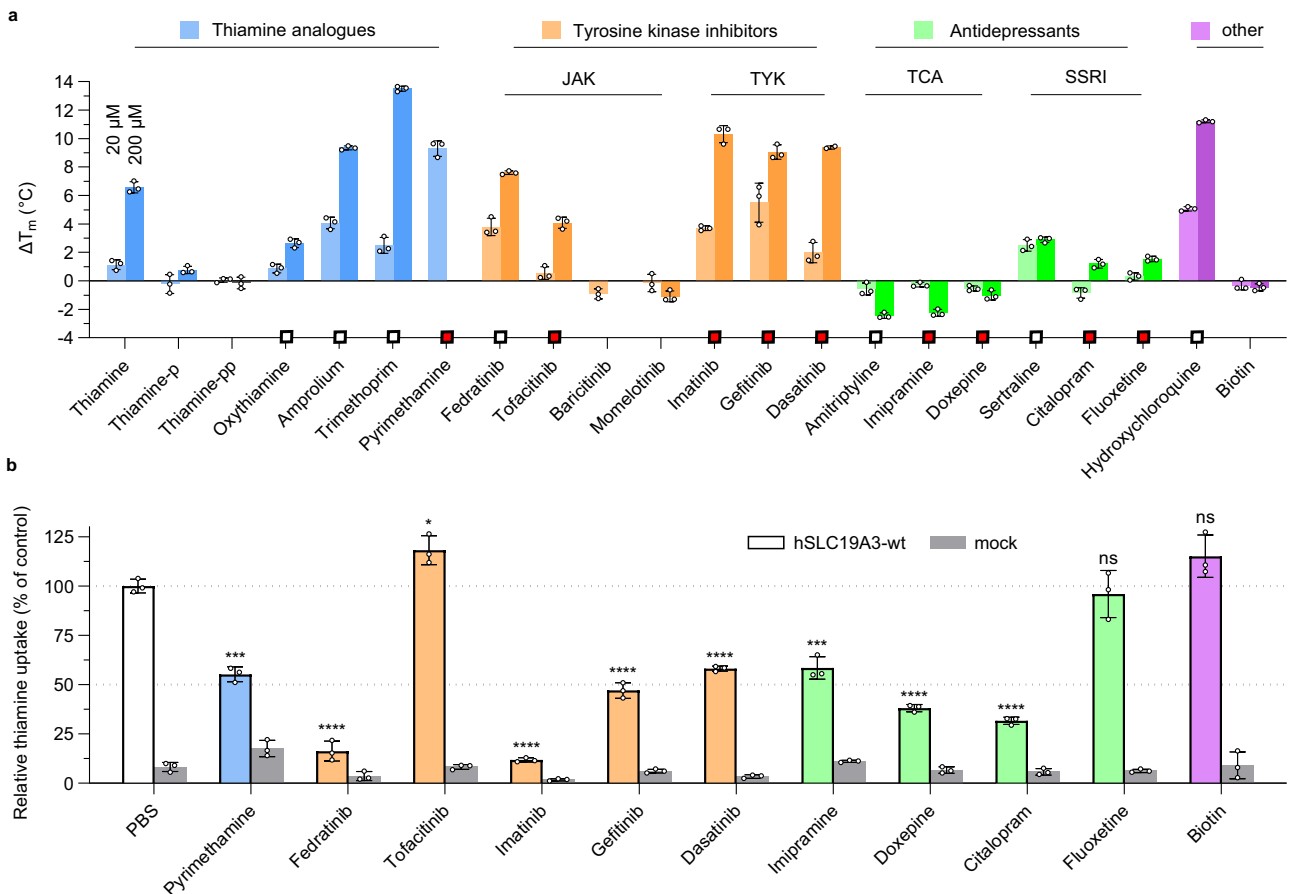

**Fig. 3 | Drug interactions of hSLC19A3. a** Thermal shift screen of known and potential thiamine uptake inhibitors (TUIs). Thermal shift assays were performed in three independent experiments ($n = 3$) in the presence of 20 µM and 200 µM of the respective compound and the stabilisation effect compared to the apo stability of hSLC19A3. The melting temperature was determined as the inflection point of the thermal unfolding curve of the protein (cf. Fig. 1d). Wherever possible, the ratio of the fluorescence at 350 nm and 330 nm (F350/F330) was selected as a readout for the thermal unfolding curve. The bar plot shows the mean ± s.d. of the measured thermal shifts. Due to intrinsic fluorescence of the following compounds, the fluorescence trace at 350 nm (F350) only was used as a readout: oxythiamine, trimethoprim, pyrimethamine, tofacitinib, imatinib, gefitinib, dasatinib, sertraline,

fluoxetine and hydroxychloroquine. The two compounds, pyrimethamine and baricitinib, interfere at 200 µM too strongly with the fluorescence detection to provide an unambiguous readout and were omitted in this dataset. The squares indicate identified TUIs. White squares indicate already known TUIs, and red squares highlight hSLC19A3-inderactions identified in this study. **b** Cellular thiamine-uptake and its inhibition by 200 µM of selected drugs. Shown is the relative uptake of deuterated thiamine (thiamine-d3, normalised to PBS-only control) by Expi293F™ cells overexpressing hSLC19A3-wt (indicated in colour), compared with mock transfected cells (indicated in grey). Three independent measurements were performed ($n = 3$). The data are shown as mean ± s.d. (ns: $p \geq 0.5$, *$p \leq 0.05$, **$p \leq 0.01$, ***$p \leq 0.001$, ****$p \leq 0.0001$, Student's $t$ test, two-tailed).

(Supplementary Figs. 18b, c and 19a, d). This part of TM7 is involved in the formation of the substrate binding site and the extracellular gate through Asn297, Gln300 and Ile301 (Fig. 2c, d, Supplementary Fig. 17). Structural changes in this region might explain the loss of affinity and transport activity of the Thr422Ala mutant.

The cryo-EM structures determined in this study additionally provide hints on the disease-causing molecular effects of other BTBGD mutations. Trp59Arg, Trp94Arg, Tyr113His, and Glu320Lys would be directly affecting thiamine binding by placing a positive charge in the substrate binding pocket of hSLC19A3 (Fig. 2e, Supplementary Fig. 18c). Binding data on the previously mentioned Glu32Lys mutant illustrate that the insertion of a positive charge in this region can lead to a strong loss of affinity for thiamine (>10-fold, Fig. 2f, Supplementary Fig. 4). This would likely render the transporter unreceptive for physiological thiamine concentrations, which is the possible cause for the pathogenicity of these mutations.

### Drug interactions of SLC19A3

Previous studies identified a broad spectrum of FDA-approved drugs that act as thiamine uptake inhibitors (TUIs) by blocking hSLC19A3[7,10,11]. This has important clinical implications, as many of these drugs are

widely prescribed over extended periods of time. Among these drugs are antidepressants, such as sertraline (Zoloft® and others) and amitriptyline, the antiparasitic hydroxychloroquine and the Janus-kinase (JAK) inhibitor fedratinib (Inrebic®). The inhibition of SLC19A3 can lead to organism-wide and tissue-specific thiamine deficiencies. In the latter case, the deficiencies would not be reflected in serum thiamine levels and remain undetected in standard diagnostics. As BTBGD-causing mutations and the example of SLC19A3 knock-out mice illustrate, this can lead to the decline of entire cell populations in vital organs, particularly in the brain[11,49]. Studying recombinantly purified hSLC19A3 in thermal shift assays, we explored the interaction space of the transporter with six known TUIs and eleven pharmacologically related drugs (Fig. 3). The corresponding screen confirmed physical binding of known inhibitors to hSLC19A3, and further led to the identification of nine previously unknown drug interactions, of which seven show inhibitor function (Fig. 3). To begin with, several drugs that are structurally related to thiamine were probed. The known hSLC19A3 inhibitors amprolium and trimethoprim strongly stabilised the transporter against heat unfolding in a concentration-dependent manner. This effect was exceeded by the antiparasitic pyrimethamine tested here (Fig. 3a). Even at a concentration of 20 µM, this compound led to a thermostabilisation

of $9.3 \pm 0.5$ °C and reduced thiamine-uptake through hSLC19A3 by about 50% in cell-based assays.

Next, representative JAK inhibitors were analysed. Fedratinib, which is known to be a potent inhibitor of hSLC19A3, also showed high-affinity binding to the transporter in the thermal shift assays ($K_{d,app} = 1.0 \pm 0.2\,\mu M$, Supplementary Fig. 21) and strongly suppressed the thiamine transport activity of hSLC19A3 (Fig. 3b). The interaction of hSLC19A3 with three other JAK inhibitors, tofacitinib (Xeljanz®), baricitinib (Olumiant®) and momelotinib (Ojjaara®) was tested as well. These compounds are functionally similar to, but structurally distinct from fedratinib. Tofacitinib is also stabilising hSLC19A3, but shows no inhibitory effect on thiamine transport (Fig. 3a). Baricitinib and momelotinib did not stabilise hSLC19A3 and thus likely do not interact with the transporter. This agrees with previous cellular uptake inhibition studies of JAK inhibitors[7].

Another important drug class that was tested were tyrosine kinase inhibitors, represented by imatinib (Gleevec®), gefitinib (Iressa®) and dasatinib (Sprycel®). They strongly stabilized hSLC19A3 with apparent binding affinities of $4.4 \pm 0.5\,\mu M$, $116 \pm 36\,nM$ and $7.5 \pm 1.5\,\mu M$, respectively (Supplementary Fig. 21). All tested tyrosine kinase inhibitors interfered with thiamine-uptake through hSLC19A3, with imatinib reaching an inhibitory effect comparable to the one of fedratinib (Fig. 3b). When comparing fedratinib and the tyrosine kinase inhibitors on a chemical level, they share a central aminopyrimidine moiety (Supplementary Fig. 21). This moiety is also at the core of the interaction of thiamine within the substrate binding site (Fig. 2e) and is thus potentially important for the interaction of the tyrosine kinase inhibitors with hSLC19A3, as already suggested by Giacomini and co-workers[7].

All of the tested tricyclic antidepressants (TCA), including the known inhibitor amitriptyline and the ones assessed here, imipramine and doxepin, interact with hSLC19A3 and inhibit its thiamine transport activity (Fig. 3). Interestingly, TCAs actually destabilize the transporter in a concentration-dependent manner, while compounds of other classes were generally stabilising. In the group of selective serotonin reuptake inhibitors (SSRI), sertraline stabilises hSLC19A3 already at low micromolar concentrations ($\Delta T_m = 2.5 \pm 0.4$ °C, Fig. 3), which is in agreement with existing data on cellular thiamine uptake inhibition[10]. The compounds citalopram (Celexa® and others) and fluoxetine (Prozac® and others) tested here also interact with hSLC19A3, but only citalopram interferes with thiamine uptake (Fig. 3). The last analysed compound was hydroxychloroquine (Fig. 3). This drug strongly stabilises hSLC19A3 and binds to the transporter with high-affinity in the nanomolar range ($K_{d,app} = 170 \pm 55\,nM$, Supplementary Fig. 21). This agrees with the previously reported high-affinity inhibition of hSLC19A3 by hydroxychloroquine[10]. In contrast to the known and the hSLC19A3 inhibitors identified here, biotin does not interact with the transporter (Fig. 3b), in agreement with our and previously published cellular uptake data[25] (Fig. 3b).

### Structural basis for thiamine uptake inhibition

The known thiamine uptake inhibitors (TUIs), as well as the ones identified here, are structurally diverse (Supplementary Fig. 21). Their chemical heterogeneity presents a challenge for the development of a universal ligand-based pharmacophore model[7]. To elucidate how these compounds, bind to hSLC19A3, we determined cryo-EM structures of the transporter in complex with three known transport inhibitors: fedratinib, amprolium and hydroxychloroquine, (Fig. 4a–c). They represent structurally and functionally different drug classes. Cryo-EM structures were solved using either Nb3.7 or Nb3.3 as structural fiducials, both of which stabilise the transporter in the inward-open state. The cryo-EM maps of the transporter-drug complexes were resolved to 3.0-3.7 Å, reaching resolutions between 2.5 and 3.2 Å in the core of the protein (Supplementary Figs. 22–24). All of the analysed inhibitors bind orthosterically in the substrate binding site of the transporter (Fig. 4). From a structural point of view, the binding

of these compounds to hSLC19A3 is determined by at least three key factors:

(1) Intercalation of aromatic rings in the aromatic clamp: Our cryo-EM data show that the aromatic rings of all three structurally resolved inhibitors form direct π-π stacking interactions with Tyr113 and to a lesser extent with Phe56 and Trp59 (Fig. 4a–c). This seems to be a universally important feature of the interaction of TUIs with hSLC19A3, as most known and all ligands identified here comprise at least one aromatic ring. This interaction might be particularly important for the binding of compounds that otherwise lack compatible polar contacts. This concerns specifically the tested antidepressants amitriptyline, imipramine, doxepin, sertraline and citalopram (Fig. 3a).

(2) Electrostatic compatibility with the polar contact points presented by the side chains of Glu32, Glu110 and Asn297: Fedratinib, amprolium and hydroxychloroquine provide hydrogen bond donors and acceptors that pair spatially well with the polar residues of the substrate binding site (Fig. 4a–c). All three inhibitors are within hydrogen bonding distance to Glu110. As mentioned above, this residue might be protonated, allowing Glu110 to act simultaneously as a hydrogen bond donor and acceptor. This would agree with the observed interactions of fedratinib and amprolium with this residue (Fig. 4a). Another important polar residue of the substrate binding site is Asn297. Its side chain carbonyl oxygen contributes to the electrostatically negative interaction surface of the transporter in general, and with 2.8 Å, it is within hydrogen bonding distance with the primary amine of amprolium (Fig. 4c). The coordination capability of Glu320 is essential for high-affinity binding of thiamine, as shown in Fig. 2f and Supplementary Fig. 19c. For inhibitor binding, however, this residue seems to play no major role, as no interactions between its side chain and the inhibitors could be observed (Fig. 4a–c). Glu32, in contrast, is in clear hydrogen bond distance with one of the secondary amines of fedratinib (2.8 Å, Fig. 4b).

(3) Insertion of a lipophilic moiety in the hydrophobic pocket formed by Thr93, Trp94, Leu97, Met106 and Val109: The structurally resolved inhibitors all engage in this type of interaction and protrude an apolar substituent into the hydrophobic pocket (Fig. 4a–c), namely, a methyl group (fedratinib), a propyl group (amprolium) and a chlorine (hydroxychloroquine). This interaction is likely beneficial for high affinity binding to hSLC19A3. A structurally corresponding methyl group can, for example, also be found in the chemical structures of the inhibitors imatinib and dasatinib, identified here (Supplementary Fig. 21).

## Discussion

Our structural and biophysical data reveal key aspects of the transport cycle and drug interactions of hSLC19A3. The transporter moves in a rocker-switch motion over the course of its transport cycle, creating a moving barrier around its substrate (Fig. 2). Our data support a model where thiamine is first recognised via the NTD in the outward-open hSLC19A3. Once the transporter changes to its inward-open conformation, thiamine rearranges to form a molecular bridge between the NTD and the CTD, before it dissociates into the cytoplasm. The binding of thiamine follows a key-in-lock mechanism, with almost no structural differences between apo and substrate-bound structures of the transporter ($C_\alpha$ RMSD of 0.30–0.62 Å; Supplementary Fig. 11). In our work, we identified the key residues mediating thiamine recognition in hSLC19A3. Several of these are mutated in severe neurometabolic diseases (BTBGD and WLE, Supplementary Fig. 18). Regarding the transport mode of hSLC19A3, it is still not clear whether the transporter operates in a uniporter fashion or requires co-substrates to drive thiamine uptake. Energetically, a uniporter mechanism is plausible. The membrane potential in combination with the concentration gradient of thiamine might be sufficient for uptake of this organic cation across the plasma membrane. Our data highlight that thiamine is not able to bind to hSLC19A3 once it is phosphorylated (Supplementary Fig. 1a,b). With this, the exit route for the vitamin via

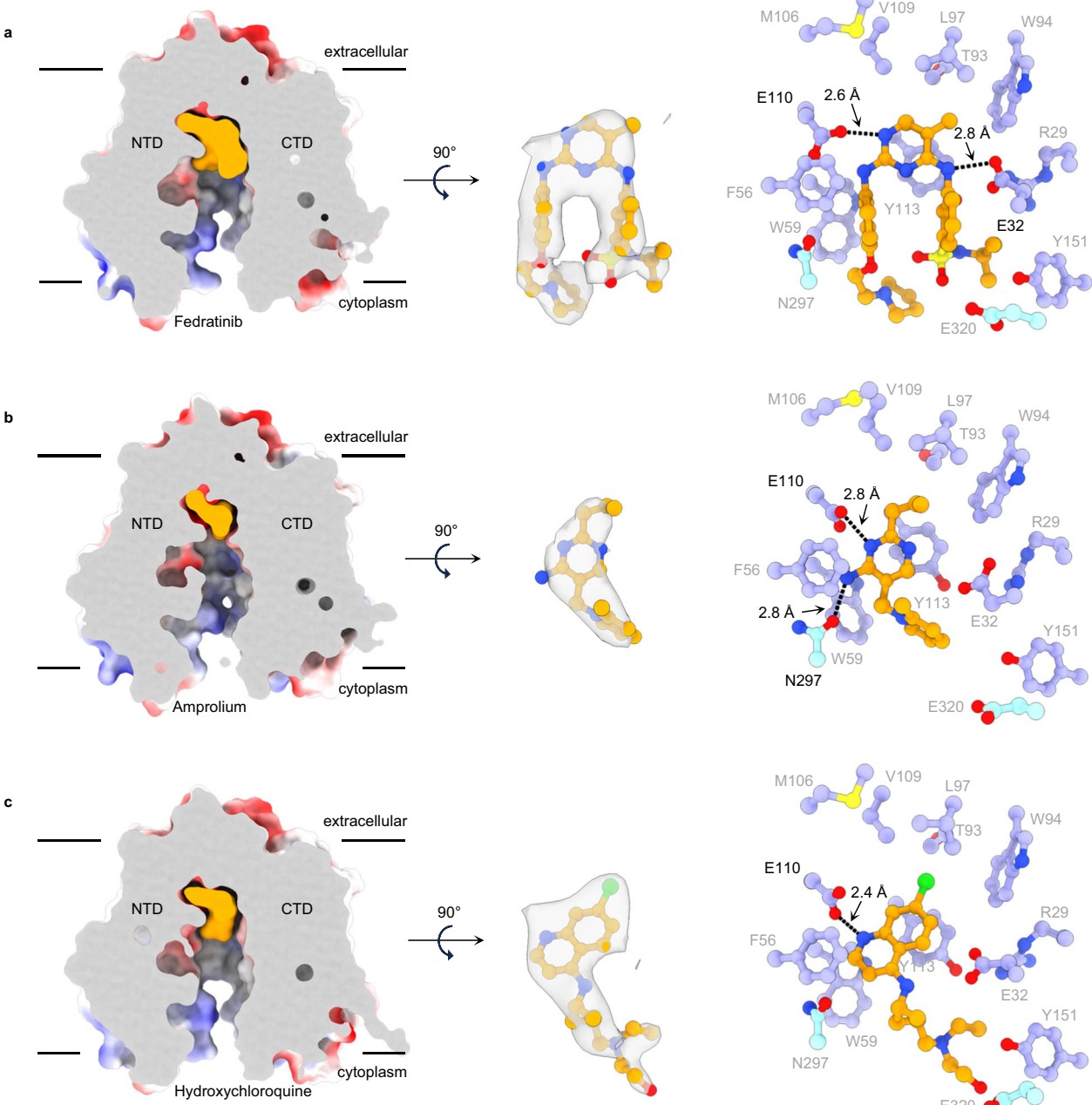

**Fig. 4 | Cryo-EM structures of hSLC19A3 in complex with high-affinity inhibitors. a–c** Bound inhibitors (illustrated in orange) were observed to occupy the same substrate binding site and engage with similar residues as thiamine. Shown on the left are cross sections of the electrostatic surface representations of the ligand-bound structures, with the ligand coloured in orange. The densities for the different compounds are shown in the centre of the respective panels (ChimeraX contour levels/Q-scores for fedratinib: 0.7/0.47, amprolium: 0.15/0.53, hydroxychloroquine: 0.47/0.75); density within 2.5 Å of the atom coordinates is shown. On the right, the coordination of the compounds in the substrate binding site is depicted. Residues of the NTD are shown in purple, the ones of the CTD in cyan and the compound in orange. Black dashes indicate hydrogen bonds (cut-off at 3.0 Å)[34].

hSLC19A3 is blocked and shifts the transport equilibrium in favour of thiamine import (Fig. 1a).

A coupling of the transport mechanism of hSLC19A3 to sodium ions was not observed in a previous study[25]. There are conflicting data about the role of protons for the activity of hSLC19A3. While some authors find evidence for proton-dependent anti- or symport mechanisms[8,50], others see no link between the uptake of thiamine and proton gradients[24]. We provide structure-based predictions, suggesting that the binding site residue Glu320 might undergo a strong $pK_a$ shift (from neutral to acidic), as it forms a salt bridge with Lys380 upon the transition of hSLC19A3 from the outward-open to the inward-open

state. This could potentially connect the uptake of thiamine to the translocation of protons across the membrane. However, functional studies, ideally in the controlled environment of liposome-based transport assays are needed to asses this hypothesis.

hSLC19A3 interacts with many commonly used medical compounds. This has implications for the adjustment and monitoring of existing treatment plans, as well as the development of new drugs[10,11]. In this work, we identified seven drugs that inhibit thiamine uptake via hSLC19A3. Among the most notable ones are the tyrosine kinase inhibitors imatinib, gefitinib and dasatinib (Fig. 3). While these in vitro findings still need to be validated in vivo, our study highlights the

importance of screening drug interactions of vitamin transporters like hSLC19A3. One aspect we did not address in the present study is the mode of inhibition of hSLC19A3 by the different drugs. For fedratinib and trimethoprim, it is known that they are xenobiotic substrates of the transporter[7], and consequently inhibit thiamine uptake by acting as competing substrates. In this context it seems plausible that hSLC19A3 might also represent a transport route for existing and prospective medical compounds. In particular, it might be an interesting entry gate for drugs to cross the blood-brain barrier[7,51]. Our structural data provide a basis for the rational design of drugs to prevent the side effects of thiamine uptake inhibition or to use hSLC19A3 for tissue-specific drug delivery[51]. The other high-affinity thiamine transporter, hSLC19A2, shares high sequence identity (48%)[26] and, based on AF2 predictions[42], high structural conservation with hSLC19A3 (Supplementary Fig. 14). We speculate that many of our findings also apply to hSLC19A2. It seems likely that this transporter follows a similar transport mechanism. Since the substrate binding site is well conserved, most inhibitors of hSLC19A3 are expected to bind to its closest relative.

During the revision of our manuscript another study by Dang and co-workers[36] was published, describing the structure of hSLC19A3 bound to thiamine and fedratinib in the inward-open state. Furthermore, Qu and colleagues released a preprint on structural work of hSLC19A3 and hSLC19A2[37]. Both studies are in line with the data and conclusions presented here and complement each other. Our study presents a structural framework of thiamine transport and drug interactions of hSLC19A3. We provide structural snapshots of its transport cycle and detailed molecular insights into the binding of its substrate and high-affinity inhibitors to the transporter. This work can serve as a basis for studies of the transport mechanism of human MFS transporters and can guide future structure-based drug design approaches.

## Methods

### Cloning, expression and purification of hSLC19A3

The full-length, wildtype hSLC19A3 (Uniprot accession number: Q9BZV2) was cloned from cDNA (Source BioScience) into a modified pXLG vector[52]. In this vector, hSLC19A3 is fused with a C-terminal human rhinovirus 3 C (HRV-3C) protease cleavage site (LEVLFQGP), followed by a short flexible linker (SSG) and a Twin-Strep-tag for affinity purification. All constructs, including mutants were generated using SLiCE cloning[53]. The corresponding construct and primer sequences are listed in the Supplementary Data file. PCR reactions were performed using an in-house purified Phusion DNA polymerase, provided in a 2×Phusion reaction mix (40 mM Tris-HCl, pH8.9, 4 mM MgCl$_2$, 120 mM KCl, 20 mM (NH$_4$)$_2$SO$_4$, 0.02 mM EDTA, 0.2 % TritonX-100, 8 % Glycerol, 0.005 % Xylene Cyanol FF, 0.05 % Orange G, 0.4 mM dNTPs, 0.04 U/µl Phusion Polymerase). The transporter constructs were transiently transfected in Expi293F™ cells using PEI MAX® as a transfection agent (ratio 1:2 (w/w) of DNA to PEI MAX®) and grown for 96 h post-transfection in FreeStyle™ medium, at 37 °C, 8% (v/v) CO$_2$, and 270 rpm[54]. Cells were harvested at 3000×g, washed with 1×PBS (pH7.4) and subsequently frozen at -70°C. After thawing at room temperature, five grams of cell pellet were resuspended in 25 mL of solubilisation buffer (1×PBS, pH7.4, 200 mM NaCl, 5% glycerol (v/v), 1% LMNG (w/v), 0.1% CHS (w/v), 1×EDTA-free protease inhibitors (Roche), 0.5 mM TCEP, 20 U/mL DNase I, 2.5 U avidin) in a 50 mL tube and incubated on a shaking platform for 1 h at 4 °C. Cellular debris was removed via centrifugation for 30 min at 35,000×g at 4 °C. The supernatant was loaded onto 1 mL StrepTactin resin and incubated for 1 h at 4 °C. Afterwards, the resin was washed with 4×20 column volumes of StrepA buffer (20 mM Tris-HCl, pH 7.4, 350 mM NaCl, 5% glycerol (v/v), 0.5 mM TCEP, 0.02% LMNG (w/v), 0.002% CHS (w/v)). The target proteins were eluted in 5 column volumes StrepB buffer (20 mM Tris-HCl, pH 7.4, 150 mM NaCl, 0.002% LMNG, 0.0002% CHS, 10 mM desthiobiotin). The eluted protein was concentrated to 0.5 mL and further purified by size-exclusion chromatography (SEC) in SEC

buffer (20 mM Tris-HCl, pH7.4, 150 mM NaCl, 0.002% LMNG (w/v), 0.0002% CHS (w/v)) on a Superdex S200 10/300 Increase column (Cytiva). Biotinylation of Avi-tagged protein for nanobody selection was performed by adding 120 µg/mL of BirA, 5 mM ATP, 50 µM biotin, 0.5 mM TCEP and 40 µg/mL 3C protease to the concentrated and desalted (PD Miditrap G-25 (Cytiva), in SEC buffer) StrepB elution. The mixture was then incubated at 4 °C overnight. Subsequently, the transporter was separated from the enzymes and the small molecule additives by SEC as described above.

### Thermal stability measurements and binding affinity determination

The thermal stability of the hSLC19A3 constructs and their binding affinities for small molecules were determined by nanoDSF, using a Prometheus NT.48 (NanoTemper Technologies). For sample preparation, the protein was diluted to a final concentration of 4 µM in SEC buffer and supplied with the respective compounds in a concentration range from 2 µM to 2 mM. Before measurement, the samples were incubated for 30 min at room temperature. Melting scans were performed at a temperature range from 20 to 95 °C with a temperature increment of 1 °C per minute. Thermal unfolding curves were recorded by measuring the ratio of the tryptophan-fluorescence at 350 nm (F350) and 330 nm (F330). The protein melting temperature ($T_m$) of a given sample was determined as the inflection point of the thermal unfolding curves, or the local maximum of its first derivative. The measured curves were analysed using the manufacturer's software (PR.ThermControl, v 2.1.2, NanoTemper Technologies). Based on the determined $T_m$ values, ligand induced thermal shifts ($\Delta T_m$) were determined as the difference between the melting temperature of the ligand-bound state ($T_m$) and the apo state ($T_m^{apo}$). For the determination of apparent binding affinities at 25 °C, the $T_m$ values recorded from dilution series of the respective ligands were used to fit the following equation:

$$\frac{\triangle T_m}{T_m^{apo}}(L) = \frac{-RT_{std}}{E_{a1}} * \ln\left(\frac{K_{d,app}}{K_{d,app} + L}\right) \tag{1}$$

were L is the ligand concentration, R is the universal gas constant, $T_{std}$ is the chosen standard temperature of 298.15 K (25 °C), $E_{a1}$ is the activation energy for the unfolding of the apo state.

As R, $T_{std}$ and L are known and the $\Delta T_m$ and $T_m^{apo}$ are experimentally determined, non-linear curve fitting can be used to estimate $E_{a1}$ and $K_{d,app}$. The fitting was performed using symfit (v. 0.5.3), as described in Kotov et al.[55], based on theoretical considerations formulated in Hall, 2019[56].

### Generation of nanobodies

To generate nanobodies against the human thiamine transporter SLC19A3, the glycosylation-free mutant hSLC19A3-gf was injected into a llama (*Glama lama*) on a weekly basis over the course of six weeks as described elsewhere[28]. Following mRNA extraction and cDNA library preparation from peripheral blood mononuclear cells of the immunized llamas, VHH-fragments were cloned into the pMESy4[28] vector, which provides the insert with an N-terminal PelB signal peptide for periplasmic expression, and a C-terminal His$_6$-tag, followed by a Myc peptide. Subsequently, two rounds of phage display were performed to enrich for specific binders. From the enriched libraries, nanobody (VHH) families were identified by sequencing of 192 colonies. Eventually, three unique nanobodies, Nb3.3, Nb3.4 and Nb3.7 were identified and expressed. Binding to the transporters was confirmed using biolayer interferometry (Supplementary Fig. 2). For the production of nanobodies, transformed *E. coli* WK6 cells were grown in 1 L TB medium, containing 0.1% glucose, at 37 °C, and 165 rpm. Overexpression of the nanobodies was induced with 1 mM IPTG at an OD of 0.7, where after the cells were grown for 18 h at 22 °C, and 165 rpm. After pelleting

the cells by centrifugation for 15 min at 7000 rpm, periplasmic extraction was performed by resuspending the pellet in 200 mL periplasmic extraction buffer (20% sucrose (w/v), 50 mM Tris-HCl, pH7.4, 0.5 mM EDTA, pH8.0, 1×EDTA-free protease inhibitors (Roche), 0.5 μg/mL lysozyme), and stirring the suspension for 30 min at 4 °C. Subsequently, the periplasmic extract was incubated with 1 mL Ni-NTA resin for 1 hour at 4 °C. The resin was then washed with 4×20 mL of wash buffer (20 mM Tris-HCl, pH7.4, 500 mM NaCl, 20 mM imidazole, pH7.4). The nanobodies were afterwards eluted from the resin with 5 mL of elution buffer (20 mM Tris-HCl, pH7.4, 500 mM NaCl, 400 mM imidazole, pH7.4). The eluted protein was concentrated to 0.5 mL and further purified by SEC using a Superdex S75 10/300 Increase column (Cytiva) in TBS buffer (20 mM Tris-HCl, pH7.4, 150 mM NaCl).

## Biolayer interferometry (BLI)
Binding between hSLC19A3 and the nanobodies Nb3.3, Nb3.4 and Nb3.7 was measured by biolayer interferometry (BLI) using an Octet RED96 system (FortéBio). The nanobodies were immobilised at 300 nM via their C-terminal 6×His-tag on Octet® HIS1K Biosensors, pre-equilibrated in BLI buffer (20 mM Tris-HCl, pH7.4, 150 mM NaCl, 0.02% LMNG (w/v), 0.002% CHS (w/v), 0.1% BSA (w/v), 0.5 mM thiamine). After a baseline step of 60 s, the biosensors were dipped into BLI buffer containing hSLC19A3-gf at concentrations ranging from 25-800 nM for 240 s to allow for the binding of the transporter to the immobilised nanobodies. Afterwards, the sensors were transferred into fresh BLI buffer and the dissociation was followed over the course of 600 s. BLI experiments were performed at 22 °C. Data were reference-subtracted and aligned in the Octet Data Analysis software v10.0 (FortéBio). Binding affinities were determined using a 1:1 binding model and the maximum response values as readout.

## Mass spectrometry-based cellular thiamine-uptake assays
The uptake of deuterated thiamine (thiamine-d3) was measured in hSLC19A3-wt and mock transfected cells in using LC-MS/MS. For this assay, Expi293F™ cells were transfected as described above and incubated for 48 h at 37 °C, 8% (v/v) CO$_2$, and 270 rpm prior to the experiment. The cells were then pelleted for 5 min at 250×$g$ and resuspended in 1×PBS (pH 7.4) at a cell density of 2 million cells per mL. The cell suspension was subsequently transferred to a 96-deep well plate in three technical replicates per condition ($n$ = 3, 1 mL per well, corresponding to 2 million cells per condition). 200 μM of the selected compounds were added to the respective wells and the cells were incubated for 1 min on a shaking platform. Subsequently, 2 μM of thiamine-d3 were added and the plate was incubated for another 5 min on a shaking platform. Afterwards, the cells were pelleted for 5 min at 500×$g$ and washed 3× with 1 mL 1×PBS (pH7.4), before transferring them to a fresh 96-deep well plate. The cell pellets were stored at −70 °C. For the LC-MS/MS analysis, the samples were extracted by adding 500 μL of a mixture of H$_2$O:MeOH:ACN (1:1:1, v/v), containing 5% (v/v) formic acid. After vortexing and ultrasonication in a water bath for 5 min at 4 °C, samples were incubated for 20 min at −20 °C. Ultimately, samples were centrifuged at 15,000×$g$ and 4 °C for 10 min using a 5415 R microcentrifuge (Eppendorf, Hamburg, Germany). The supernatants were then transferred for LC-MS analysis, which was initiated within one hour after sample preparation. LC-MS/MS analysis was performed on a Vanquish UHPLC system coupled to an Orbitrap Exploris 240 high-resolution mass spectrometer (Thermo Fisher Scientific, MA, USA) in positive ESI (electrospray ionization) mode. Chromatographic separation was carried out on an Atlantis Premier BEH Z-HILIC column (Waters, MA, USA; 2.1 mm × 100 mm, 1.7 μm) at a flow rate of 0.25 mL/min. The mobile phase consisted of water:acetonitrile (9:1, v/v; mobile phase phase A) and acetonitrile:water (9:1, v/v; mobile phase B), which were modified with a total buffer concentration of 10 mM ammonium formate. The aqueous portion of each mobile phase was adjusted to pH 3.0 via addition of formic acid. The

following gradient (8 min total run time including re-equilibration) was applied (time [min]/%B): 0/90, 3/85, 3.5/60, 4/60, 5/90, 8/90. Column temperature was maintained at 40 °C, the autosampler was set to 4 °C and sample injection volume was set to 3 μL. Analytes were recorded via a full scan with a mass resolving power of 120,000 over a mass range from 60 to 900 $m/z$ (scan time: 100 ms, RF lens: 70%). MS/MS fragment spectra were recorded via targeted product ion scans for Thiamine ([M]$^+$, $m/z$ = 265.1118) and Thiamine-d$_3$ ([M]$^+$, $m/z$ = 268.1306) at a resolving power of 15,000, stepped collision energies [%]: 20/35/50, and total cycle time of 3 s. Ion source parameters were set to the following values: spray voltage: 3500 V, sheath gas: 30 psi, auxiliary gas: 5 psi, sweep gas: 0 psi, ion transfer tube temperature: 350 °C, vaporizer temperature: 300 °C.

All experimental samples were measured in a randomized manner. Pooled quality control (QC) samples were prepared by mixing equal aliquots from each processed sample. Multiple QCs were injected at the beginning of the analysis in order to equilibrate the analytical system. A QC sample was analyzed after every 6[th] experimental sample to monitor instrument performance throughout the sequence. For determination of background signals and subsequent background subtraction, an additional processed blank sample was recorded. Data was processed using TraceFinder 5.1 and raw peak area data was exported for relative metabolite quantification.

## Cryo-EM sample preparation and data acquisition
For cryo-EM, hSLC19A3 was purified in LMNG/CHS as described above. hSLC19A3-wt was prepared in complex with Nb3.3 and Nb3.4, while hSLC19A3-gf was used in conjunction with Nb3.7. After concentrating the transporter to 60 μM, the sample was supplied with a 1.5× molar excess of nanobody and diluted to a total concentration of 30 μM of hSLC19A3 in SEC buffer. Compounds were added during the dilution step at the following concentrations: 500 μM thiamine, 250 μM fedratinib, 300 μM amprolium or 200 μM hydroxychloroquine,. The samples were subsequently incubated for 4–18 h on ice. 3.6 μL of sample were applied to glow discharged holey carbon film grids (either Quantifoil 300 mesh, Au, R1.2/1.3, or R2/1, or C-flat, 200 mesh, Cu, 2/1) for 15 s at 10 °C and 100% humidity in a Vitrobot Mark IV (Thermo Fisher Scientific). Blotting was performed for 3.5 s at blot force -5 and with a drain time of 0.5 s. The grids were plunged into liquid ethane/propane and transferred to liquid nitrogen for storage. Most data were collected in counting mode on a Titan Krios G3 (FEI) operating at 300 kV with a BioQuantum imaging filter (Gatan) and a K3 direct detection camera (Gatan). Data were collected in fast acquisition mode by aberration-free image shift (AFIS) at 105,000× magnification, a physical pixel size of 0.83 or 0.85 Å, a slit width of 20 eV and a total dose of 45-60 e$^-$/Å$^2$ (dose rate of -15 e$^-$/px/s, exposure time of -2.4 s; for details, see Supplementary Table 1). The nominal defocus was for most datasets was set to the range of -0.8 μM to -1.6 μm. The dataset for the hydroxychloroquine-bound hSLC19A3 was collected on a Titan Krios G4 equipped with a cold-FEG, a Falcon 4i electron detector camera, and a SelectrisX imaging filter with its slit width set to 10 eV. The data were acquired at a nominal magnification of 215,000×, corresponding to a pixel size of 0.572 Å.

## Cryo-EM data processing
Processing of the acquired micrographs was performed using cryoSPARC (v. 4.4)[57]. Briefly, the collected movies were motion corrected using patch motion correction. After patch CTF estimation, exposures with a CTF resolution <3.5 Å, a relative ice thickness <1.05 and a total frame motion distance <30 pixels were selected for further processing. Particle picking was performed using template picker. The corresponding templates were generated from blob-picking (100 Å particle diameter) a small subset of the data (200 - 600 micrographs), which was subjected to 2D classification, ab-initio reconstruction and non-uniform refinement (NU-refinement). After template-based particle

picking of the entire dataset, protein-containing and well-aligning particles were enriched using several rounds of 2D classification. The selected particles were then used in a sequence of ab-initio reconstruction and NU-refinement to create a first higher resolution map, usually in the resolution range of 3.5-4.0 Å. This refined map was then, together with randomly generated volumes, used for several rounds of heterogeneous refinement to retrieve more good particles and remove junk particles. The resulting map was then further refined using NU-refinement. The corresponding particles were subsequently subjected to local motion correction and re-extracted with a larger box size of 384 pixels (~326 Å). After another round of NU-refinement, the resulting density map was locally refined using a mask covering the transporter-nanobody complex. This was followed by local resolution estimation and local filtering to finalise the maps. For more details, see Supplementary Fig. 6. Q-scores were determined using UCSF Chimera (v.1.18) and the MapQ plugin (v.1.9.14), developed by Grigore Pintilie[58].

## Model building and refinement
For starting models, AlphaFold2 predictions of hSLC19A3 and the nanobodies were used[42]. These were relaxed into the corresponding EM density maps using ISOLDE[59]. Ligands were built and parametrised using the eLBOW package within PHENIX[60] and integrated in the protein structures using Coot (v. 0.9.8.1)[61]. The resulting models were subsequently refined iteratively in PHENIX and Coot. Figures were prepared using UCSF ChimeraX (v. 1.6.1)[59].

All reagents generated in this study are available from the Lead Contact with a completed Materials Transfer Agreement.

## Reporting summary
Further information on research design is available in the Nature Portfolio Reporting Summary linked to this article.

## Data availability
The EM data and fitted models for hSLC19A3 generated in this study have been deposited in the Protein Data Bank (PDB) and the Electron Microscopy Data Bank (EMDB) under the following accession codes: hSLC19A3-wt:Nb3.4-apo [https://doi.org/10.2210/pdb8S4U/pdb], [https://www.ebi.ac.uk/pdbe/entry/emdb/EMD-19716]; hSLC19A3-wt:Nb3.4:thiamine [https://doi.org/10.2210/pdb8S5U/pdb], [https://www.ebi.ac.uk/pdbe/entry/emdb/EMD-19750]; hSLC19A3-wt:Nb3.3-apo [https://doi.org/10.2210/pdb9G5K/pdb], [https://www.ebi.ac.uk/pdbe/entry/emdb/EMD-51088]; hSLC19A3-gf:Nb3.7:thiamine [https://doi.org/10.2210/pdb8S61/pdb], [https://www.ebi.ac.uk/pdbe/entry/emdb/EMD-19754]; hSLC19A3-gf:Nb3.7:Fedratinib [https://doi.org/10.2210/pdb8S5W/pdb], [https://www.ebi.ac.uk/pdbe/entry/emdb/EMD-19752]; hSLC19A3-gf:Nb3.7:Amprolium [https://doi.org/10.2210/pdb8S62/pdb], [https://www.ebi.ac.uk/pdbe/entry/emdb/EMD-19755], hSLC19A3-gf:Nb3.7: Hydroxychloroquine [https://doi.org/10.2210/pdb8S5Z/pdb], [https://www.ebi.ac.uk/pdbe/entry/emdb/EMD-19753]. Other data generated in this study are provided in the Supplementary Data 1 file and Supplementary Movie 1, Supplementary Movie 2, and Supplementary Movie 3. Source Data are provided with this paper as source data file. Source data are provided with this paper.

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

## Acknowledgements

We thank the Sample Preparation and Characterization facility at EMBL (Hamburg, Germany) and the team of the cryo-EM Facility at CSSB for their support, technical assistance and advice. We want to acknowl-edge Tânia Custódio, Grzegorz Chojnowski and all group members for fruitful discussions and continuous support and feedback on the pro-ject. Part of this work was performed at the CryoEM Facility at CSSB, supported by the UHH and DFG (grant numbers INST 152/772-1 | 152/774-1 | 152/775-1 | 152/776-1 | 152/777-1 FUGG). We further acknowledge the access and services provided by the Imaging Centre at the Eur-opean Molecular Biology Laboratory (EMBL IC) in Heidelberg, gener-ously supported by the Boehringer Ingelheim Foundation. In this context, we want to thank Joseph Bartho for his help with the cryo-EM data acquisitions at the Imaging Centre. We want to acknowledge the support of the EMBL Metabolomics Core Facility (MCF), and in parti-cular the help of Bernhard Drotleff, in the acquisition and analysis of liquid chromatography-mass spectrometry data. Nanobody discovery was realised through the Nanobodies4Instruct centre and the support by Instruct-ERIC (PID: 23688), which is part of the European Strategy Forum on Research Infrastructures (ESFRI). We want to thank in parti-cular Alison Pirro Lundqvist and Eva Beke from the Steyaert Lab at VUB for their technical assistance during nanobody discovery. The research-stay of F.G. at VUB was supported by an EMBO Scientific Exchange Grand (Nr. 10251).

## Author contributions

F.G. conceptualised the project and experiments, cloned and expressed transporter constructs and nanobodies, performed the nanobody discovery, performed and analysed nanoDSF and BLI experiments, collected and processed all cryo-EM data and prepared the manuscript. L.S. and A.F. cloned and expressed transporter constructs, as well as nanobodies, and collected biophysical data. K.E.J.J. and S.M. helped with experiment design, as well as cryo-EM data processing, structure refinement and manuscript writing. J.S and E.P. immunised llamas and generated nanobody libraries for the discovery process. C.L. was responsible for the overall project administration, experiment design, funding acquisition, and for reviewing and editing of the manuscript.

## Funding

## Competing interests

The authors declare no competing interests.
