## [Peer Review File · Nature Communications]

Structural basis of thiamine transport and drug recognition by SLC19A3REVIEWER COMMENTS

Reviewer #1 (Remarks to the Author):

Gabriel et al report several structures of human SLC19A3 stabilized in multiple different conformations in the presence of nanobody binders, including a structure for the outward facing state of the transporter at resolutions ranging from 2.4-3.8 Angstroms and in the presence of substrates and inhibitors. The structural data are complemented by nanoDSF and thermostability measurements for wild type and point mutations of binding site residues and disease-causing mutations. The importance of this transporter and high quality of the structural work and insights gleaned, including the determination of the outward facing structure, that likely extend to other members of the SLC19A family, make this manuscript in principle suitable for publication in nature communications. However, the following points should be considered:

1. While the structural work including nanobody discovery and conformational trapping is One weakness of the manuscript is the lack of direct binding analysis or thiamin translocation assays. Do the authors have conclusive proof that nanobody binding doesn't alter thiamin/inhibitor binding kinetics?
2. Similar to point 1 above, thermal stability shift assays alone are insufficient to make conclusive statements on the effect of mutations on ligand binding and transport. In the absence of direct binding or transport assays the authors should include this caveat in their discussion on ligand binding and apparent binding constants.
3. Minor point: The authors should include EM density for Thiamin and all associated ligands in their Supp. Fig7, Supp. Fig10, Supp. Fig11, Supp. Fig12, Supp. Fig13.

Reviewer #2 (Remarks to the Author):

Gabriel et al have determined the first cryo-EM structures of the thiamine transporter SLC19A3. Using nanobodies raised to the purified protein, they determined structures in two main states, the outward-open state and the inward-open state, both in the absence and presence of thiamine. The structures confirm that the thiamine transporter has the canonical fold of members of the Major Facilitator Superfamily and that the mechanism of transport follows the similar path for other members of the family such as LacY. The authors identify residues that are involved in the binding of thiamine and show that the conformation of thiamine changes during the transition from the outward-open to the inward-open conformation. A number of genetic diseases arise from mutations in the SLC19A3 gene and the authors show that many of the mutations affect residues that bind directly to thiamine or are involved during the structural transition between in the outward-open and inward-open states.

Substrate binding to many transporters will increase their thermostability, and the authors show this to be true for the thiamine transporter as well. Using this assay, a number of known uptake inhibitors were tested and also shown to alter the thermostability of the transporter. A number of related drugs were also tested and many showed the potential to interact with the thiamine transporter as observed by a change in thermostability. The structures of the thiamine transporter bound to four known inhibitors of thiamine uptake were then determined in the inward-open

conformation, showing that they bind in the same cleft as thiamine and interact with similar amino acid residues. This is of interest in the pharmaceutical industry as they may cause side effects through inhibiting thiamine uptake.

The manuscript is on the whole well written although clarity could be improved, particularly in the figures.

Major issues

1. In the structures of the apo states of the transporter, there is unexplained weak density in the position where thiamine binds. This should be shown and discussed in the manuscript. Could this be something picked up from expression in the cells or from the buffers during purification? It may be worth trying focussed classification on the region around the substrate binding pocket to see if this separates into different subclasses with different occupancies.
2. In modelling all the different structures, in some cases the rotamer of a residue cannot be defined from the density and different rotamers have been used in different structures. You may wish to be consistent in assigning the rotamer in these instances. This is particularly pertinent for residues in the binding pocket where it is now uncertain whether the different rotamers are due to different modelling strategies or due to the different substrates bound.
3. For each of the structures, please show a figure of the substrate/inhibitor in the binding cleft with density for the surrounding side chains as well as for the substrate/inhibitor. Please do the same for the apo structures.
4. The authors discuss at some length the protonation of Glu110 and its role in the binding of thiamine and there is a blurring between what is fact from the structures and what is hypothesis. It is not possible to observe hydrogen atoms in the structures as the resolution is insufficient. This must be stated clearly if the authors decide to keep this discussion in the manuscript. I think the evidence here is rather weak for protonation given that the pKa of Glu110 is only predicted to lie within the range of 6.4-7.5 and that the pH of blood is between 7.35-7.45 and the pH in the gut is even more variable (duodenum, 5.6-8.0, small intestine 7.2-7.5 and colon 7.9-8.5; I do not know what area of intestine the thiamine transport is found). More clarity is definitely required as at pH 7.5 Glu110 could be 90% protonated (pKa 6.5) or 50% protonated (pKa 7.5). It is also unclear how the pKa changes in the presence of thiamine (I think the pKa for Glu110 was determined in the absence of substrate, but no specific mention is made in the text). Is there any transport data to show that pH affects the rate of thiamine uptake (in the absence of a proton gradient and under conditions of constant membrane potential)? If there is, please cite this as justification for your hypothesis. If not, then the discussion could be more accurately centred around the strong polar interaction made by Glu110. Hydrogen atoms must be removed from any figures showing structures (Supplementary Fig. 1d,e; Supplementary Fig. 23b,d; and any I have missed). Where hydrogen atoms have been inserted into schematic representations (Supplementary Fig. 1a; Fig. 4e) it must be stated in the figure legend that 'hydrogen atoms were not observed in the cryo-EM structures and that their position is hypothetical'. One other point, if Glu110 is protonated, you will not have delocalised electrons between the two oxygen atoms as shown in the figures, but the charge localised on the unprotonated oxygen.
5. In Figure 3a, I am a little perplexed about what defines an inhibitor. There are of course clear examples where there is an increase in thermostability at 20 μ M and a greater increase at 200 μ M, and then further work to determine an apparent Kd. However, doxepine is a known inhibitor and there is a slight decrease in thermostability in the thermostability assay. Similar results were obtained with momelotinib, but this was classified as not being an inhibitor. The phraseology in the

discussion of these data needs to be careful in assessing whether a compound is an inhibitor or not; a negative result does not prove that a compound is NOT an inhibitor in the absence of data from other assays e.g. inhibition of thiamine uptake into cells.

6. It would be helpful for the reader to have a direct comparison between the different conformations of thiamine by superposing the ligand structures after protein alignment. It would also be good to have a similar alignment of all the inhibitors with thiamine after alignment of the transporter structures.

Minor issues

1. In substrate-bound structures the substrate is assigned a number e.g. chainA/601, but in the apo structure a lipid is also assigned as 601. This is potentially very confusing and should be changed.
2. Line 84: define percent homology rather than just saying 'closely related'
3. Line 89-90: The sentence is ambiguous, as it could refer to not deleting SLC19A2 nor OCT1 in the SLC19A3 knock-out strain. It is probably clearer to delete 'but neither SLC19A2 nor OCT1' and put this in a new sentence e.g. 'Deletion of either SLC19A2 or OCT1 does not affect thiamine levels in the body'. Also define that these experiments were performed in mice.
4. Line 134: The reference to Supplementary Fig.1 refers only to panel b. There is a lot of unrelated material in this figure that is cited later in the text, so this needs re-organising.
5. Line 148: the reference to Supplementary Fig.5-13 is incorrect as it also refers to the drug-bound structures. For clarity, just state that all the structural information for the apo structures, thiamine-bound and drug bound structures may be found in Supplementary Fig.5-13 and Supplementary Table 1.
6. Line 325: 'theses' should be 'these'
7. Line 455: insert 'potential' before 'high-affinity binders'
8. Line 815: please state that this cartoon was derived from hydropathy analysis of the amino acid sequence.
9. Throughout the manuscript: In the figure legends, it is often stated 'side view' or 'top view' when referring to models. This is unscientific. Please use terminology that relates the view to the membrane eg. extracellular view, viewed in the membrane plane etc
10. Line 851: state where the primary data are shown.
11. Line 868: state the number of technical replicates, experimental replicates, and errors.
12. Supplementary Fig.1c: make the stick models clearer
13. Supplementary Fig. 2 legend, line 7; why 'presumably'? Delete?.
14. Supplementary Fig. 3. Make the densities a darker shade of grey to make them visible.
15. Supplementary Fig. 4: Why do none of the curves reach saturation? For Y151F the highest concentration used is 100x the apparent Kd, so the curve should be flat. This suggests that background has not been appropriately taken into account. Please discuss in the methods the origin of the background and why it cannot be removed. The thiamine binding affinity for N297A should not be cited as this is just background binding.
16. Supplementary Fig.5-13. The resolution range used in the maps showing local resolution of the cryo-EM density should be tailored for each structure so that a range of colours are seen and not just all blue which does not convey any information. Please use rainbow colouration to give more information.
17. Supplementary Fig. 14 : in panels a & b, please use darker colours so it is possible to see the structures and show a key on the panels to say which structure is which. In c & d, please use

separate colours for all four structures as it appears that only two structures are being compared. In panel b, the structures are referred to as 'rocker-switch'; what does this mean? Please use a standard nomenclature throughout the manuscript for the structures.

Reviewer #3 (Remarks to the Author):

Gabriel et al. reported the structures of SLC19A3 in complexes with its native substrate thiamine as well as several drugs. It is indeed commendable that the authors could obtain the structures of this significant vitamin transporter in both the inward-open and outward-open states by using specific nanobodies. On this basis, the authors further discussed the substrate and drug recognition mechanisms of this transporter and also proposed a potential substrate transport mechanism. Overall, this study is very solid, systematic, and will be of great significance to the field. I would like to recommend publishing in Nature Communications. However, there are some suggestions that I would like to make for the authors to consider before the manuscript is finally accepted.

1. As mentioned in the manuscript, it is still under debating about the transport mode of SLC19A3. It might be uniporter, symporter, or antiporter as reported by different studies. Here the authors proposed a thiamine-proton co-transport model in Supplementary Fig. 18. This model should be carefully deliberated. It is reported that lower extracellular pH could inhibit the thiamine transport activity of SLC19A3 by several labs (Rajgopal et al. BBA, 2001, PMID: 11731220; Yamashiro et al. JBC, 2020, PMID: 33008889; Dang et al. Cell Res., 2024, PMID: 38503960). However, the model here is contradictory to these experimental results. Please provide more functional data to support this model. The major evidence the author provided is the pKa shift of E320 in different conformations. This calculation really relies on the local environment or the accuracy of the structural models. For example, the side chain orientation of K380 could severely affect the result. Thus, there might be some ambiguity in the calculation of pKa. We cannot simply draw any conclusion solely based on this calculation.

2. It seems E32 is also important for the substrate binding as mutating it to lysine decreased the affinity of thiamine by 10-fold (Fig. 2f). Is it possible that this acidic residue is also involved in proton coupling just like E320? What is the pKa of this residue in different functional states?

3. E110 is vital for thiamine binding as shown in Fig. 2c-e, why did not the authors verify its effect on thiamine binding by mutagenesis studies in Fig. 2f as other residues?

4. As Dang et al. reported the SLC19A3 structures recently (Dang et al. Cell Res., 2024, PMID: 38503960), it is better to compare the current structures with their studies. Are the structural models and substrate transport mechanisms consistent with their work?

5. In this paper the only functional assay is the thermal stability measurements. This assay could indicate the binding affinity of the SLC19A3 variants with different compounds to some extent, but could not fully reflect the transport activity. The author might consider to verify the substrate transport activity of some key mutants to further support their conclusions, such as the disease-causing mutations.

6. The EM densities of hydroxychloroquine and amitriptyline could not fit their structures very well

as shown in Fig. 4a and 4d. MD simulations might help assign the correct poses of these compounds if possible.

Minor issues:

1. Not only the EM densities of the substrates and drugs, but also the densities of the residues around need to be shown in the corresponding Figures or as Supplementary Figures.
2. SLC19A3 was written as SLC15A3 in lines 560 and 565 of the Methods section.
3. Please further refine the structural models to eliminate the rotamer outliers.
4. All the model-map FSC curves are required in Supplementary Figures to show the model-map agreement.

Reviewer Comments

Reviewer #1 (Remarks to the Author):

Gabriel et al report several structures of human SLC19A3 stabilized in multiple different conformations in the presence of nanobody binders, including a structure for the outward facing state of the transporter at resolutions ranging from 2.4-3.8 Angstroms and in the presence of substrates and inhibitors. The structural data are complemented by nanoDSF and thermostability measurements for wild type and point mutations of binding site residues and disease-causing mutations. The importance of this transporter and high quality of the structural work and insights gleaned, including the determination of the outward facing structure, that likely extend to other members of the SLC19A family, make this manuscript in principle suitable for publication in nature communications. However, the following points should be considered:

1. While the structural work including nanobody discovery and conformational trapping is One weakness of the manuscript is the lack of direct binding analysis or thiamin translocation assays. Do the authors have conclusive proof that nanobody binding doesn't alter thiamin/inhibitor binding kinetics?

Response:

We fully agree on the first point and now provide direct thiamine-uptake data (mass spectrometry-based) in the revised manuscript (see Fig. 3). This is further complemented by work recently published on hSLC19A3 by Dang et al., 2024 and Qu et al., 2024, which is consistent with our findings. As to the second point raised here: we currently do not have a technique established to directly measure the binding kinetics of small molecule ligands to the transporter. With thermal shift assays, we probe the overall thermodynamics of the binding equilibrium. Based on our structural data we show that binding of nanobodies does not significantly impact the structure of hSLC19A3, nor its general ability to bind thiamine. For the inward-open state, we don't see any discernible differences between the EM maps/structure models determined in the presence of Nb3.3 vs. the ones determined with Nb3.7. Furthermore, the nanobody-bound structures are consistent with low resolution ($\sim 7 \text{ \AA}$ resolution), fiducial-free structures that we determined for inward-open hSLC19A3 (these data are not shown in the manuscript). For the outward-open state we cannot make these comparisons within our own data. But structures of hSLC19A3 in complex with an outward-open state stabilising Fab fragment, recently published by Qu et al., 2024, are highly similar to our nanobody-bound structures.

2. Similar to point 1 above, thermal stability shift assays alone are insufficient to make conclusive statements on the effect of mutations on ligand binding and transport. In the absence of direct binding or transport assays the authors should include this caveat in their discussion on ligand binding and apparent binding constants.

Response:

We agree on this point. We now added transport data to the manuscript (Fig. 3) and also refer to the other two publications/preprints describing the structure, and in particular the transport activity of hSLC19A3 (Dang et al., 2024, Qu et al., 2024).

3. Minor point: The authors should include EM density for Thiamin and all associated ligands in their Supp. Fig7, Supp. Fig10, Supp. Fig11, Supp. Fig12, Supp. Fig13.

Response:

We've added the EM densities for the ligands correspondingly.

Reviewer #2 (Remarks to the Author):

Gabriel et al have determined the first cryo-EM structures of the thiamine transporter SLC19A3. Using nanobodies raised to the purified protein, they determined structures in two main states, the outward-open state and the inward-open state, both in the absence and presence of thiamine. The structures confirm that the thiamine transporter has the canonical fold of members of the Major Facilitator Superfamily and that the mechanism of transport follows the similar path for other members of the family such as LacY. The authors identify residues that are involved in the binding of thiamine and show that the conformation of thiamine changes during the transition from the outward-open to the inward-open conformation. A number of genetic diseases arise from mutations in the SLC19A3 gene and the authors show that many of the mutations affect residues that bind directly to thiamine or are involved during the structural transition between in the outward-open and inward-open states.

Substrate binding to many transporters will increase their thermostability, and the authors show this to be true for the thiamine transporter as well. Using this assay, a number of known uptake inhibitors were tested and also shown to alter the thermostability of the transporter. A number of related drugs were also tested and many showed the potential to interact with the thiamine transporter as observed by a change in thermostability. The structures of the thiamine transporter bound to four known inhibitors of thiamine uptake were then determined in the inward-open conformation, showing that they bind in the same cleft as thiamine and interact with similar amino acid residues. This is of interest in the pharmaceutical industry as they may cause side effects through inhibiting thiamine uptake.

The manuscript is on the whole well written although clarity could be improved, particularly in the figures.

Major issues

1. In the structures of the apo states of the transporter, there is unexplained weak density in the position where thiamine binds. This should be shown and discussed in the manuscript. Could this be something picked up from expression in the cells or from the buffers during purification? It may be worth trying focused classification on the region around the substrate binding pocket to see if this separates into different subclasses with different occupancies.

Response:

We have added a discussion point on this topic in the results section and show this more clearly in Supplementary Fig. 13. Our work highlights that the substrate binding site of SLC19A3 is rather promiscuous and binds various kinds of small molecules. The extra density observed in the apo structures therefore likely originates from different small molecules picked up from the medium/cells/buffer. We also followed the recommendation of the reviewer and tried various tools for focused classification: Using a focus mask on the entire transporter in 3D classification results in fluctuations of the weak densities observed in the substrate binding site of the apo state. However, we think that 3D classification in cryo-EM is not (yet) suitable to detect such small mass differences (<500 Da). We would in fact treat this with a lot of caution, as the classification is likely to sort noise of the particle images into different 3D classes and thus create artefacts in what we would interpret as ligand density. A classification focused on just the substrate binding site integrates too little mass for alignment and leads to a complete distortion of the sidechain and possible ligand density within the focus region.

2. In modelling all the different structures, in some cases the rotamer of a residue cannot be defined from the density and different rotamers have been used in different structures. You may wish to be consistent in assigning the rotamer in these instances. This is particularly pertinent for residues in the binding pocket where it is now uncertain whether the different rotamers are due to different modelling strategies or due to the different substrates bound.

Response:

This is a valid point and we have curated the structure models correspondingly. In cases of uncertainty, we rely on the rotamer of the respectively highest resolution structure to inform our choices for lower resolution maps. E.g. the orientation of Trp94 in the drug-bound states was eventually determined from the density for this residue in the fedratinib-bound state of SLC19A3; and we found that that rotamer orientation was in good agreement with the lower resolution maps of e.g. amprolium and modelled it correspondingly.

3. For each of the structures, please show a figure of the substrate/inhibitor in the binding cleft with density for the surrounding side chains as well as for the substrate/inhibitor. Please do the same for the apo structures.

Response:

This information has been added in Supplementary Fig. 13 & 25.

4. The authors discuss at some length the protonation of Glu110 and its role in the binding of thiamine and there is a blurring between what is fact from the structures and what is hypothesis. It is not possible to observe hydrogen atoms in the structures as the resolution is insufficient. This must be stated clearly if the authors decide to keep this discussion in the manuscript. I think the evidence here is rather weak for protonation given that the pKa of Glu110 is only predicted to lie within the range of 6.4-7.5 and that the pH of blood is between 7.35-7.45 and the pH in the gut is even more variable (duodenum, 5.6-8.0, small intestine 7.2-7.5 and colon 7.9-8.5; I do not know what area of intestine the thiamine transport is found). More clarity is definitely required as at pH 7.5 Glu110 could be 90% protonated (pKa 6.5) or 50% protonated (pKa 7.5). It is also unclear how the pKa changes in the presence of thiamine (I think the pKa for Glu110 was determined in the absence of substrate, but no specific mention is made in the text). Is there any transport data to show that pH affects the rate of thiamine uptake (in the absence of a proton gradient and under conditions of constant membrane potential)? If there is, please cite this as justification for your hypothesis. If not, then the discussion could be more accurately centred around the strong polar interaction made by Glu110. Hydrogen atoms must be removed from any figures showing structures (Supplementary Fig. 1d,e; Supplementary Fig. 23b,d; and any I have missed). Where hydrogen atoms have been inserted into schematic representations (Supplementary Fig. 1a; Fig. 4e) it must be stated in the figure legend that 'hydrogen atoms were not observed in the cryo-EM structures and that their position is hypothetical'. One other point, if Glu110 is protonated, you will not have delocalised electrons between the two oxygen atoms as shown in the figures, but the charge localised on the unprotonated oxygen.

Response:

We agree. We have been a bit blunt with putting these hypotheses forward. We have adjusted our wording to make clear that the protonation of Glu110 is a possibility, but can by no means be "proven" by our data. We have also adapted the concerned figures, and figure legends. We think it is valuable to point at the potential involvement of Glu110 and Glu320 in proton-coupling. But we limit our elaboration now to what we know and only include a short section on possible protonation in the discussion section. The corresponding Supplementary Fig. proposing a coupling mechanism has been omitted in the revised manuscript to avoid confusion.

5. In Figure 3a, I am a little perplexed about what defines an inhibitor. There are of course clear examples where there is an increase in thermostability at 20 uM and a greater increase at 200 uM, and then further work to determine an apparent Kd. However, doxepine is a known inhibitor and there is a slight decrease in thermostability in the thermostability assay. Similar results were obtained with momelotinib, but this was classified as not being an inhibitor. The phraseology in

the discussion of these data needs to be careful in assessing whether a compound is an inhibitor or not; a negative result does not prove that a compound is NOT an inhibitor in the absence of data from other assays e.g. inhibition of thiamine uptake into cells.

Response:

We agree on this point. We've tried to be careful with the wording, but we understand that it can be misleading. In the revised manuscript, we have included the results of cellular thiamine-uptake inhibition assays. These are based on the quantification of deuterated thiamine (thiamine-d3) by mass spectrometry (Fig. 4). The corresponding data show that seven, out of the nine hits identified in the thermal shift assays, interfere with thiamine uptake via SLC19A3. In light of these results, we propose the following wording for the paper: We state that compounds that induce a melting temperature shift of the transporter "interact" with SLC19A3. And the compounds that significantly reduce the uptake of thiamine in the cell-based assays "inhibit" thiamine transport by SLC19A3.

6. It would be helpful for the reader to have a direct comparison between the different conformations of thiamine by superposing the ligand structures after protein alignment. It would also be good to have a similar alignment of all the inhibitors with thiamine after alignment of the transporter structures.

Response:

Our approach was to yield a good comparison of the structures through juxtaposition of protein-aligned structures in the main figures and some superpositions in the supplementary figures. In our view, and after trying it, a superposition of the ligand-bound structures does not necessarily paint a clearer picture. We hope that the revised figures are clear enough in that regard.

Minor issues

1. In substrate-bound structures the substrate is assigned a number e.g. chainA/601, but in the apo structure a lipid is also assigned as 601. This is potentially very confusing and should be changed.

Response:

We've revised all structure models and corrected for that error. The numbering should be uniform now.

2. Line 84: define percent homology rather than just saying 'closely related'.

Response:

We've expressed this now in percent of sequence identity and similarity, to have a concrete measure.

3. Line 89-90: The sentence is ambiguous, as it could refer to not deleting SLC19A2 nor OCT1 in the SLC19A3 knock-out strain. It is probably clearer to delete 'but neither SLC19A2 nor OCT1' and put this in a new sentence e.g. 'Deletion of either SLC19A2 or OCT1 does not affect thiamine levels in the body'. Also define that these experiments were performed in mice.

Response:

We've clarified this now in the text.

4. Line 134: The reference to Supplementary Fig.1 refers only to panel b. There is a lot of unrelated material in this figure that is cited later in the text, so this needs re-organising.

Response:

We have reorganized the figure to be clearer.

5. Line 148: the reference to Supplementary Fig.5-13 is incorrect as it also refers to the drug-bound structures. For clarity, just state that all the structural information for the apo structures, thiamine-bound and drug bound structures may be found in Supplementary Fig.5-13 and Supplementary Table 1.

Response:

We agree that the referencing in that line is ambiguous. We've adjusted it correspondingly.

6. Line 325: 'theses' should be 'these'.

Response:

This has been changed.

7. Line 455: insert 'potential' before 'high-affinity binders'

Response:

This has been added.

8. Line 815: please state that this cartoon was derived from hydrophathy analysis of the amino acid sequence.

Response:

The cartoon was in its details derived from the experimental structures. We've stated that now in the text.

9. Throughout the manuscript: In the figure legends, it is often stated 'side view' or 'top view' when referring to models. This is unscientific. Please use terminology that relates the view to the membrane eg. extracellular view, viewed in the membrane plane etc

Response:

This has been adapted correspondingly in the revised manuscript.

10. Line 851: state where the primary data are shown.

Response:

This information has been added.

11. Line 868: state the number of technical replicates, experimental replicates, and errors.

Response:

This information has been added.

12. Supplementary Fig.1c: make the stick models clearer.

Response:

The figure has been changed as suggested

13. Supplementary Fig. 2 legend, line 7; why 'presumably'? Delete?

Response:

This has been changed.

14. Supplementary Fig. 3. Make the densities a darker shade of grey to make them visible.

Response:

This has been changed.

15. Supplementary Fig. 4: Why do none of the curves reach saturation? For Y151F the highest concentration used is 100x the apparent K_d , so the curve should be flat. This suggests that background has not been appropriately taken into account. Please discuss in the methods the origin of the background and why it cannot be removed. The thiamine binding affinity for N297A should not be cited as this is just background binding.

Response:

These are not classical binding experiments, like e.g. radioligand binding. The curves show compound-induced thermostabilisation of the target protein. A full plateauing of the curve is not expected due to the logarithmic dependency of the thermal shift on the compound concentration, as described in the methods section (for further examples cf. e.g. Kotov et al., 2023, Cell Reports or Jungnickel et al., 2024, Nature Cell Biology). Likewise, the concept of background cannot be directly transferred from classical binding experiments. In the case of the mutant N297A, thiamine induces a smaller, yet significant and concentration-dependent thermostabilisation of SLC19A3. We include a corresponding negative control and raw data for clarity in Supplementary Fig. 5.

16. Supplementary Fig.5-13. The resolution range used in the maps showing local resolution of the cryo-EM density should be tailored for each structure so that a range of colours are seen and not just all blue which does not convey any information. Please use rainbow colouration to give more information.

Response:

Setting a fixed local resolution range from 2.5-4.5 Å was a conscious decision on our side with the rationale of making the local resolutions of the different maps comparable amongst each other. That comes at the cost of less ideal coloration for some maps, like the one of hSLC19A3:Nb3.3-apo. But we think that the above mentioned point weighs more in this context. We chose the blue-white scale to make the figures fully inclusive for colour blind readers and avoid the skewing effect that rainbow palettes have on the human perception of scales (cf. Crameri et al., 2020).

17. Supplementary Fig. 14: in panels a & b, please use darker colours so it is possible to see the structures and show a key on the panels to say which structure is which. In c & d, please use separate colours for all four structures as it appears that only two structures are being compared. In panel b, the structures are referred to as 'rocker-switch'; what does this mean? Please use a standard nomenclature throughout the manuscript for the structures.

Response:

We have made the colour in panels a & b darker and have added a legend that should allow for an easy interpretation of the structure comparison. In panels c & d, we would like to stick to the chosen colours to highlight the difference between apo and thiamine-bound structures. But we've added a clearer legend. In panel e, we have adapted the nomenclature to be consistent with the rest of the manuscript and easier to understand.

Reviewer #3 (Remarks to the Author):

Gabriel et al. reported the structures of SLC19A3 in complexes with its native substrate thiamine as well as several drugs. It is indeed commendable that the authors could obtain the structures of this significant vitamin transporter in both the inward-open and outward-open states by using specific nanobodies. On this basis, the authors further discussed the substrate and drug recognition mechanisms of this transporter and also proposed a potential substrate transport mechanism. Overall, this study is very solid, systematic, and will be of great significance to the field. I would like to recommend publishing in Nature Communications. However, there are some suggestions that I would like to make for the authors to consider before the manuscript is finally accepted.

1. As mentioned in the manuscript, it is still under debating about the transport mode of SLC19A3. It might be uniporter, symporter, or antiporter as reported by different studies. Here the authors proposed a thiamine-proton co-transport model in Supplementary Fig. 18. This model should be carefully deliberated. It is reported that lower extracellular pH could inhibit the thiamine transport activity of SLC19A3 by several labs (Rajgopal et al. BBA, 2001, PMID: 11731220; Yamashiro et al. JBC, 2020, PMID: 33008889; Dang et al. Cell Res., 2024, PMID: 38503960). However, the model here is contradictory to these experimental results. Please provide more functional data to support this model. The major evidence the author provided is the pKa shift of E320 in different

conformations. This calculation really relies on the local environment or the accuracy of the structural models. For example, the side chain orientation of K380 could severely affect the result. Thus, there might be some ambiguity in the calculation of pKa. We cannot simply draw any conclusion solely based on this calculation.

Response:

As also mentioned in the response to reviewer 2, we now removed the speculative part of the transport model and base it solely on structural observations. We would like to provide more support data for the transport mode, but cell-based transport assays are insufficient to substantiate a concrete model of the transport mechanism; and SLC19A3 has so far defied our efforts to measure its activity in liposome-based assays. We've adapted the text and figures correspondingly and are more careful with the model that we propose.

2. It seems E32 is also important for the substrate binding as mutating it to lysine decreased the affinity of thiamine by 10-fold (Fig. 2f). Is it possible that this acidic residue is also involved in proton coupling just like E320? What is the pKa of this residue in different functional states?

Response:

We have reassessed our cryo-EM data with respect to the role of Glu32 in thiamine binding. In our first analysis this residue has not been attributed the significance it deserves. The reason for this was primarily our rather strict cut-off distance (3 Å) chosen for defining hydrogen bonds, which stressed the importance of the interaction of thiamine with Glu110, but left the interaction with Glu32 somewhat neglected. We've corrected this in the revised manuscript. In addition to the observed loss in binding affinity in the Glu32Lys mutant, Dang et al., 2024 and Qu et al., 2024 show that mutation of Glu32 to alanine decreases thiamine-uptake by up to 60%. These findings are discussed in the results part of our revised manuscript. As for the predicted pKa of this residue we see no state-dependent shift of the acidity of Glu32, as its coordination by Arg29 does not change dramatically. But as discussed in other parts of this rebuttal, we will be very cautious making claims about protonation and proton coupling based on structural data alone.

3. E110 is vital for thiamine binding as shown in Fig. 2c-e, why did not the authors verify its effect on thiamine binding by mutagenesis studies in Fig. 2f as other residues?

Response:

This is a fair question. The honest answer is that, at the time of manuscript submission, the mutation of this residue to alanine had repeatedly failed on the level of site-directed mutagenesis. In the meantime, we've mutated E110 to a glutamine and to a lysine residue. In the case of E110Q, we observed a drop of the apparent affinity for thiamine from ~12 μM to ~85 μM. The E110K mutant did not show thiamine-binding at all. This indicates that a mutation to glutamine is per se tolerable for thiamine binding, as it at least partly resembles the polar contacts necessary for the

coordination of the aminopyrimidine group of the vitamin. However, when the electrostatic charge of this residue is inverted, this appears to fully abolish thiamine binding to the transporter. We demonstrate this in Supplementary Fig. 5 and discuss this in the main text.

4. As Dang et al. reported the SLC19A3 structures recently (Dang et al. Cell Res., 2024, PMID: 38503960), it is better to compare the current structures with their studies. Are the structural models and substrate transport mechanisms consistent with their work?

Response:

This work was published shortly after our first submission of this manuscript. But in the meantime, we compared our results extensively to the work of Dang et al. and the recently released preprint on hSLC19A2 and hSLC19A3 by Qu and co-workers. We find that our structural data are largely in agreement with their findings. This is particularly reassuring as all three groups have used different approaches to provide hSLC19A3 with the fiducials necessary for structure determination by cryo-EM (Dang et al.: BRIL-fusion in complex with an anti-BRIL Fab fragment; Qu et al.: hSLC19A3-specific Fab fragments; in our case: hSLC19A3-specific nanobodies). There are only minor differences where we think that our slightly higher resolution data support our claims (e.g. coordination of the hydroxyethyl tail of thiamine in the inward-open state). Apart from this we think the work is nicely complementary, especially with regard to the biophysical study of mutants of hSLC19A3. While we provide binding affinity estimations, Dang et al. and Qu et al. show extensive screens of the effects of mutants on the transport activity of hSLC19A3.

5. In this paper the only functional assay is the thermal stability measurements. This assay could indicate the binding affinity of the SLC19A3 variants with different compounds to some extent, but could not fully reflect the transport activity. The author might consider to verify the substrate transport activity of some key mutants to further support their conclusions, such as the disease-causing mutations.

Response:

We fully agree. We have added the data of cell-based thiamine uptake assays to the revised manuscript. We have used these assays to focus on the inhibition of SLC19A3 by the interacting drugs identified by the thermal shift assays. By now there is a paper (Dang et al., 2024, Cell Research) and a preprint (Qu et al., 2024, Research Square) that study the effect of mutations on the transport activity of SLC19A3. We think that the results on the SLC19A3 mutants published by these labs are consistent and don't need to be repeated by us.

6. The EM densities of hydroxychloroquine and amitriptyline could not fit their structures very well as shown in Fig. 4a and 4d. MD simulations might help assign the correct poses of these compounds if possible.

Response:

We agree that the EM maps for these two ligands were ambiguous with regard to the exact poses of the drug molecules. We therefore went on and tried to achieve higher resolutions using the latest microscope (Titan Krios G4, cold-FEG, SelectrisX energy filter, Falcon 4 camera) and microscope settings adjusted for higher target resolutions (0.52 Å pixel size, 58 e⁻/px/s, 10-20k movies per dataset). In the case of hydroxychloroquine, we were able to determine the structure to a global resolution of 3.0 Å. Importantly, the density for the ligand was well-defined and allowed us to unambiguously model the pose of both the aromatic ring system and the elongated tail of the drug. Unfortunately, a new data set in the presence of amitriptyline did not lead to an improved reconstruction. Therefore, we decided to omit the tricyclic antidepressant from the structural work presented in this manuscript to avoid any confusion or misinterpretation. We think that the ambiguous pose of amitriptyline is an inherent feature of the binding mode. It lacks polar contact points that could pair with the electrostatic profile of the substrate binding site of SLC19A3. We therefore speculate that the binding to the transporter is mainly driven by hydrophobic and π - π stacking interactions with the hydrophobic pocket and the aromatic clamp. This might allow for flexible binding/several different binding poses, which could explain our findings in the EM maps. But because we have no further data to support this idea, we are not elaborating on it in the revised manuscript.

Minor issues:

1. Not only the EM densities of the substrates and drugs, but also the densities of the residues around need to be shown in the corresponding Figures or as Supplementary Figures.

Response:

We've included corresponding supplementary figures now.

2. SLC19A3 was written as SLC15A3 in lines 560 and 565 of the Methods section.

Response:

Well spotted. Thank you. This has been corrected now.

3. Please further refine the structural models to eliminate the rotamer outliers.

Response:

We've curated the models correspondingly.

4. All the model-map FSC curves are required in Supplementary Figures to show the model-map agreement.

Response:

We've included the model-map FSC curves in the supplementary figures now.

REVIEWERS' COMMENTS

Reviewer #1 (Remarks to the Author):

The authors have addressed the minor points I had raised.

Reviewer #2 (Remarks to the Author):

The authors have made the majority of changes requested, although I still disagree with there comments regarding the figures showing map resolution. Given this is not fo critical importance, they can stay as they are if the authors do not mind the reduced information content of some of them.

Reviewer #3 (Remarks to the Author):

All my concerns have beed addressed. I support the publication of this manuscript.

Reviewer Comments

Reviewer #1 (Remarks to the Author):

The authors have addressed the minor points I had raised.

Response:

Thanks a lot for the positive feedback.

Reviewer #2 (Remarks to the Author):

The authors have made the majority of changes requested, although I still disagree with there comments regarding the figures showing map resolution. Given this is not fo critical importance, they can stay as they are if the authors do not mind the reduced information content of some of them.

Response:

As stated last time, we decided to show a fixed local resolution range from 2.5-4.5 Å making the different maps comparable amongst each other. That comes at the cost of less ideal coloration for some maps, like the one of hSLC19A3:Nb3.3-apo, but we think that the abovementioned point weighs more in this context and we would like to keep the current colouring.

Reviewer #3 (Remarks to the Author):

All my concerns have beed addressed. I support the publication of this manuscript.

Response:

We appreciate the positive feedback.